# Sheared LLaMA: Accelerating Language Model Pre-training via Structured Pruning

**Mengzhou Xia[1], Tianyu Gao[1], Zhiyuan Zeng[2] , Danqi Chen[1]**
[1]Princeton Language and Intelligence,  Princeton University
[2]Department of Computer Science and Technology, Tsinghua University
{mengzhou,tianyug,danqic}@cs.princeton.edu
zengzy20@mails.tsinghua.edu.cn

## Abstract

The popularity of LLaMA (Touvron et al., 2023a;b) and other recently emerged moderate-sized large language models (LLMs) highlights the potential of building smaller yet powerful LLMs. Regardless, the cost of training such models from scratch on trillions of tokens remains high. In this work, we study structured pruning as an effective means to develop smaller LLMs from pre-trained, larger models. Our approach employs two key techniques: (1) *targeted structured pruning*, which prunes a larger model to a specified target shape by removing layers, heads, and intermediate and hidden dimensions in an end-to-end manner, and (2) *dynamic batch loading*, which dynamically updates the composition of sampled data in each training batch based on varying losses across different domains. We demonstrate the efficacy of our approach by presenting the **Sheared-LLaMA** series, pruning the LLaMA2-7B model down to 1.3B and 2.7B parameters. Sheared-LLaMA models outperform state-of-the-art open-source models of equivalent sizes, such as Pythia, INCITE, OpenLLaMA and the concurrent TinyLlama models, on a wide range of downstream and instruction tuning evaluations, while requiring only 3% of compute compared to training such models from scratch. This work provides compelling evidence that leveraging existing LLMs with structured pruning is a far more cost-effective approach for building competitive small-scale LLMs.[1]

## 1 Introduction

Large language models (LLMs) are extremely performant on a wide range of natural language tasks, but they require enormous amounts of compute to train (OpenAI, 2023; Anthropic, 2023). As such, there is growing interest in building strong moderate-sized models, such as LLaMA (Touvron et al., 2023a;b), MPT (MosaicML, 2023), and Falcon (Almazrouei et al., 2023), that allow for efficient inference and fine-tuning. These LLMs are available in varied sizes suited for different use cases, but training each individual model from scratch—even the smallest billion-parameter models—requires substantial computational resources that are cost-prohibitive for most organizations. In this work, we seek to address the following question:

> *Can we produce a smaller, general-purpose, and competitive LLM by leveraging existing*
> *pre-trained LLMs, while using much less compute than training one from scratch?*

We explore structured pruning as a means to achieve this goal. Pruning is commonly viewed as a solution for compressing task-specific models (Han et al., 2016; Li et al., 2016; Lagunas et al., 2021; Xia et al., 2022; Kurtic et al., 2023), removing redundant parameters and accelerating inference without sacrificing task performance. However, for general-purpose LLMs, pruning inevitably results in performance degradation compared to original models (Frantar & Alistarh, 2023; Sun et al., 2023; Ma et al., 2023), especially when without significant compute invested post-pruning. In this work, we use pruning as an effective approach for developing smaller yet competitive LLMs that require only a fraction of the training compute compared to training them from scratch.

---

[1]Please find our code and models at https://github.com/princeton-nlp/LLM-Shearing. We present frequently asked questions and answers in Appendix G.

We identify two key technical challenges in this problem. First, how can we decide on final pruned architectures that are strong in performance and efficient for inference? Existing structured pruning techniques for LLMs (Xia et al., 2022; Ma et al., 2023) do not specify targeted structures and lead to suboptimal pruned models in terms of performance and inference speed (Table 4 and Appendix F.2). Second, how can we continue pre-training the pruned model to reach desired performance? We observe that training using the original pre-training data leads to imbalanced rates of loss reduction across different domains, compared to when training such models from scratch. This indicates that the pruned model retains varying levels of knowledge for different domains (e.g., GitHub vs. C4) and simply using the pre-training domain proportion results in an inefficient use of data (Figure 4). To address these issues, we propose "LLM-shearing", an algorithm consisting of the following two components:

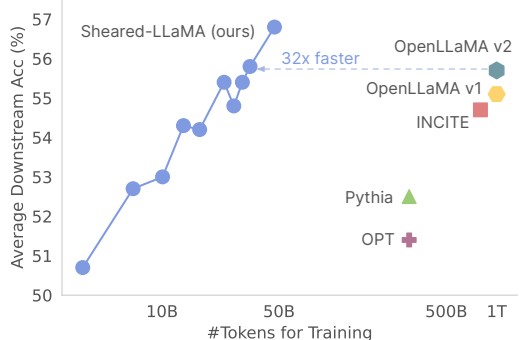

- We propose a novel pruning algorithm, dubbed *targeted structured pruning*, which prunes a source model to a specified target architecture. The target architecture is determined by leveraging the configurations of existing pre-trained models. Our pruning approach searches for substructures within the source model that maximally preserve performance while adhering to the given constraints.

- We devise a *dynamic batch loading* algorithm that loads training data from each domain in proportion to its rate of loss reduction, thereby making an efficient use of the data and accelerating the overall performance improvement.

Figure 1: Sheared-LLaMA-2.7B surpasses a series of open-source models at a similar scale and only requires 1/32 (3%) of budget to achieve on-par performance with OpenLLaMA-3B-v2.

We demonstrate the efficacy of our proposed method by pruning a LLaMA2-7B model (Touvron et al., 2023b) into two smaller LLMs: Sheared-LLaMA-1.3B and Sheared-LLaMA-2.7B. Despite using only 50 billion addtional tokens (i.e., 5% of OpenLLaMA's pre-training budget) for pruning and continued pre-training, Sheared-LLaMA-1.3B and Sheared-LLaMA-2.7B outperform other popular LLMs at similar scales, including Pythia (Biderman et al., 2023), INCITE (TogetherAI, 2023b), and OpenLLaMA (Geng & Liu, 2023), on 11 representative downstream tasks (Figure 1; commonsense, reading comprehension, and world knowledge) and instruction tuning for open-ended generation. Additionally, the downstream performance trajectory suggests that further training the pruned model with more tokens would result in even greater gains. While we only conduct experiments with up to 7B parameter models, our LLM-shearing algorithm is highly generalizable and can be extended to large language models of any size in future work.

## 2 LLM-SHEARING

Given an existing large model $\mathcal{M}_S$ (the *source* model), we study how to efficiently produce a smaller, strong model $\mathcal{M}_T$ (the *target* model). We consider this as a two-stage process: (1) Pruning $\mathcal{M}_S$ into $\mathcal{M}_T$. This reduces the number of parameters but inevitably incurs a performance drop. (2) Continue pre-training $\mathcal{M}_T$ with a standard language modeling objective to reach a target performance. While most recent efforts (Xia et al., 2022; Ma et al., 2023) focus on the former stage, we find the latter stage crucial for producing competitive general-purpose LLMs from structured pruning.

### 2.1 TARGETED STRUCTURED PRUNING

Structured pruning removes groups of model parameters to compress models and accelerate inference. However, existing structured pruning approaches often result in unconventional model configurations that deviate from popular architectures. For example, CoFiPruning (Xia et al., 2022) produces models with non-uniform layer configurations (e.g., different numbers of heads across layers), which incurs inference overhead compared to standard uniform layer configurations (Section 4.2).

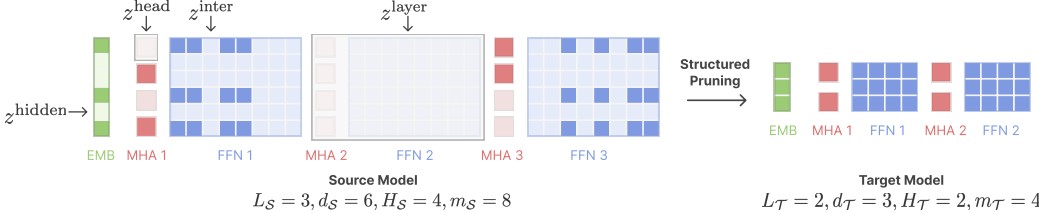

Figure 2: *Targeted structured pruning* produces a compact and dense model of a pre-specified shape. Light colors indicate pruned substructures. Masking variables $z$ are learned to control whether a substructure is pruned ($z = 0$) or retained ($z = 1$).

In this work, we aim to prune the source model into any target configuration that we specify. This goal is challenging because it requires surgically scaling down all dimensions in a transformer architecture, an endeavor that, to our knowledge, has not been accomplished before for large language models. We leverage the configurations of existing pre-trained models as the target architectures, based on the intuition that these configurations have already been well-optimized to balance model expressivity and inference efficiency. For example, we use the INCITE-Base-3B architecture (TogetherAI, 2023a) as the target structure when producing a 2.7B model.

Our method learns a set of pruning masks on model parameters at different granularities—from global ones like layers and hidden dimensions (persist across all layers), to local ones like attention heads and intermediate dimensions. Assume that the source model $\mathcal{M}_S$ has $L_S$ layers, with each layer consisting of one multi-head attention module (MHA) and one feed-forward network (FFN). $\mathcal{M}_S$ has a hidden state dimension of $d_S$, $H_S$ heads in each MHA, and an intermediate dimension of $m_S$ in each FFN. We introduce the following mask variables:

| Granularity | Layer | Hidden dimension | Head | Intermediate dimension |
|---|---|---|---|---|
| Pruning masks | $z^{\text{layer}} \in \mathbb{R}^{L_S}$ | $z^{\text{hidden}} \in \mathbb{R}^{d_S}$ | $z^{\text{head}} \in \mathbb{R}^{H_S} \ (\times L_S)$ | $z^{\text{int}} \in \mathbb{R}^{m_S} \ (\times L_S)$ |

Each mask variable controls whether the associated substructure is pruned or retained. For example, we remove a layer if its corresponding $z^{\text{layer}} = 0$. Figure 2 illustrates an example of how the pruning masks control the pruned structures.

We formulate pruning as a constrained optimization problem (Platt & Barr, 1987) where we learn pruning masks to search for a subnetwork matching a pre-specified target architecture while maximizing performance.[2] Following the $\ell_0$ regularization approach (Louizos et al., 2018), we parametrize the pruning masks to model hard concrete distributions. These distributions have support on $[0, 1]$ but concentrate their probability mass at 0 or 1, enabling discrete prune or retain decisions. While prior work usually control for a target sparsity (Wang et al., 2020; Xia et al., 2022), we use a pair of Lagrange multipliers to impose constraints on the pruned model shape directly. For example, for a target number of heads $H_T$ (and we use $L_T$, $d_T$, and $m_T$ to represent the target number of layers, hidden dimension, and intermediate dimension respectively), we have the imposed constraint on a single layer as:

$$\tilde{\mathcal{L}}^{\text{head}}(\lambda, \phi, z) = \lambda^{\text{head}} \cdot \left(\sum z^{\text{head}} - H_T\right) + \phi^{\text{head}} \cdot \left(\sum z^{\text{head}} - H_T\right)^2.$$

Similar constraints are applied to pruning other substructures. Overall, we jointly optimize the model weights and pruning masks by a min-max objective $\min_{\theta, z} \max_{\lambda, \phi} \mathcal{L}_{\text{prune}}(\theta, z, \lambda, \phi)$:

$$\mathcal{L}_{\text{prune}}(\theta, z, \lambda, \phi) = \mathcal{L}(\theta, z) + \sum_{j=1}^{L_S} \tilde{\mathcal{L}}^{\text{head}}_j + \sum_{j=1}^{L_S} \tilde{\mathcal{L}}^{\text{int}}_j + \tilde{\mathcal{L}}^{\text{layer}} + \tilde{\mathcal{L}}^{\text{hidden}},$$

where $\mathcal{L}(\theta, z)$ is the language modeling loss computed with the masked model weights. This objective will produce a pruned model with the target shape. Ideally, running this pruning algorithm on a large amount of data will directly produce a strong compact model. In practice, the pruning stage is expensive (roughly $5\times$ slower compared to standard LM training), and we find that the learned

---

[2]Please find a more detailed exposition of the algorithm in Appendix A.

---

**Algorithm 1:** Dynamic Batch Loading

---

**Require**: Training data of $k$ domains $D_1, D_2, \cdots, D_k$, validation data $D_1^{\text{val}}, D_2^{\text{val}}, \cdots, D_k^{\text{val}}$,
  initial data loading weights $w_0 \in \mathbb{R}^k$, reference loss $\ell_{\text{ref}} \in \mathbb{R}^k$, LM loss $\mathcal{L}$ or pruning loss
  $\mathcal{L}_{\text{prune}}$, training steps $T$, evaluation per $m$ steps, model parameters $\theta$ ($\theta, z, \phi, \lambda$ for pruning)

**for** $t = 1, \cdots, T$ **do**
  **if** $t \mod m = 0$ **then**
    $\ell_t[i] \leftarrow \mathcal{L}(\theta, z, D_i^{\text{val}})$ if *pruning* else $\mathcal{L}(\theta, D_i^{\text{val}})$
    $\Delta_t[i] \leftarrow \max\{\ell_t[i] - \ell_{\text{ref}}[i], 0\}$               ▷ Calculate loss difference
    $w_t \leftarrow \texttt{UpdateWeight}(w_{t-m}, \Delta_t)$       ▷ Update data loading proportion
  **end**
  Sample a batch of data $\mathcal{B}$ from $D_1, D_2, \cdots, D_k$ with proportion $w_t$;
  **if** *pruning* **then**
    Update $\theta, z, \phi, \lambda$ with $\mathcal{L}_{\text{prune}}(\theta, z, \phi, \lambda)$ on $\mathcal{B}$
  **else**
    Update $\theta$ with $\mathcal{L}(\theta, \mathcal{B})$
  **end**
**end**

**Subroutine** $\texttt{UpdateWeight}(w, \Delta)$
  $\alpha \leftarrow w \cdot \exp(\Delta)$                 ▷ Calculate the unnormalized weights
  $w \leftarrow \frac{\alpha}{\sum_i \alpha[i]}$ **return** $w$        ▷ Renormalize the data loading proportion
**return** $\theta$

---

masks often converge fast. Therefore, we only allocate a limited budget for pruning (see Table 5). Following pruning, we finalize the pruned architecture by preserving the highest-scoring components associated with the mask variables in each substructure, and continue pre-training the pruned model with the language modeling objective. We refer to this second stage as continued pre-training.

## 2.2 DYNAMIC BATCH LOADING

Continued pre-training on a large amount of data is crucial for recovering the pruned model performance. We observe a surprising finding in our preliminary experiments: continuing pre-training our pruned models on an existing pre-training dataset RedPajama (TogetherAI, 2023b; LLaMA's replicated pre-training dataset) reduces loss at different rates across domains compared to pre-training a model from scratch, which signifies an inefficient use of data.

To be more specific, we begin by fitting a *scaling function* (Hoffmann et al., 2022; details in Appendix B) on the series of LLaMA2 models for each domain. Using this function, we predict the loss of a hypothetical 1.3B LLaMA2 model if it were trained from scratch on the same data. We then compare these estimated *reference losses* to the losses of our pruned model after continued pre-training. Figure 4 (left) shows that our model's loss on GitHub is better than the reference loss, while it is significantly worse than the reference loss on C4. This observation indicates that pruning preserves a greater amount of knowledge in low-entropy and smaller domains (e.g., GitHub) compared to high-entropy and larger domains (e.g., C4). Simply reusing the original pre-training data distribution[3] results in an inefficient use of data and worse downstream performance, even if the overall loss is seemingly low, as demonstrated later in Section 4.1.

Inspired by recent work (Xie et al., 2023), we propose *dynamic batch loading*, an efficient algorithm to adjust domain proportions on the fly based on losses. The goal is to ensure the model achieves the reference loss at roughly the same time across domains. We introduce the algorithm below.

**Problem setup.** The pre-training data comprises of $k$ domains $D_1, D_2, \cdots, D_k$ and we have a held-out validation dataset for each domain, denoted as $D_i^{\text{val}}$. At each training step $t$, a proportion $w_t[i]$ of the data comes from domain $D_i$. We set a reference validation loss $\ell_{\text{ref}}(D_i)$ for each domain and train the pruned model to reach the reference loss.

---

[3]The LLaMA2 pre-training data is not public. We conducted the same analysis on LLaMA1 models and observed a similar phenomenon, indicating that this is a universal issue unrelated to specific pre-training data.

**Dynamic batch loading.** We present the full algorithm in Algorithm 1. In a sketch, for every $m$ steps, we evaluate the model to get the validation loss $\ell_t$ (step $t$) on $D^{\mathrm{val}}$, and update $w_t$ based on the difference $\Delta_t(D_i)$ between $\ell_{\mathrm{ref}}[i]$ and $\ell_t[i]$ on each domain. The update rule is exponential ascent following Xie et al. (2023),

$$\alpha_t = w_{t-m} \cdot \exp(\Delta_t); \quad w_t = \frac{\alpha_t}{\sum_i \alpha_t[i]}.$$

We apply dynamic batch loading to both the pruning stage and the continued pre-training stage. For pruning, we use the original pre-training data's domain weights as $w_0$. For continued pre-training, we use the final weights from the pruning stage as $w_0$. Dynamic batch loading is an on-the-fly solution that adjusts data proportions during training without the need for training auxiliary models. It leverages reference losses on validation sets and adjusts the weights dynamically, adding minimal overhead to standard training. This approach differs from Xie et al. (2023), which requires a complex multi-stage process to train reference and proxy models.

More broadly, dynamic batch loading can train an LLM to match any reference model's performance by using open-source pre-training datasets like RedPajama, even without knowing the reference model's exact training data.

**Choices of reference losses.** By default, we use the loss predicted by the fitted scaling function as the reference (denoted as *scaling reference*). We also experiment with an alternative where we directly use the source model's domain validation loss as the reference (denoted as *source reference*). We show in F.4 that while both variants perform well, using scaling reference leads to slightly better downstream results, especially on math and coding tasks. However, source reference is a viable alternative when a series of source models at different scales is not available.

## 3 EXPERIMENTS

### 3.1 SETUP

**Model configurations.** We use the LLaMA2-7B model (Touvron et al., 2023b) as the source model throughout all of our main experiments.[4] We then conduct structured pruning experiments to compress this model down to two smaller target sizes—2.7B and 1.3B parameters. We compare to strong pre-trained language models of similar sizes, including OPT-1.3B (Zhang et al., 2022), Pythia-1.4B (Biderman et al., 2023), TinyLlama-1.1B (Zhang et al., 2024), OPT-2.7B, Pythia-2.8B, INCITE-Base-3B (TogetherAI, 2023b), OpenLLaMA-3B-v1, and OpenLLaMA-3B-v2 (Geng & Liu, 2023). We use Pythia-1.4B and INCITE-Base-3B as the target architecture for the 1.3B and the 2.7B model respectively. Table 8 summarizes model architecture details of all these models.

**Data.** As the training data for LLaMA2 is not publicly accessible, we use RedPajama (TogetherAI, 2023b), which is a replicated pre-training dataset of the LLaMA1 models (Touvron et al., 2023a), for pruning and continued-pretraining. This dataset encompasses training data from seven domains: CommonCrawl, C4, Github, Wikipedia, Books, ArXiv, and StackExchange. We construct a held-out validation set with 2 million tokens (equivalent to 500 sequences of 4,096 tokens) for each domain. We allocate 0.4 billion tokens for the pruning phase and 50 billion tokens for the continued pre-training process. Following the conventions of LLaMA2, we maintain a sequence length of 4,096 tokens. Table 1 provides a summary of the pre-training data used by our models and the baseline models.

Table 1: A summary of pre-training datasets used by Sheared-LLaMA and other models.

| Model | Pre-training Data | #Tokens |
|---|---|---|
| LLaMA1 | LLaMA data | 1T |
| LLaMA2 | *Unknown* | 2T |
| OPT | OPT data[5] | 300B |
| Pythia | The Pile | 300B |
| INCITE-Base | RedPajama | 800B |
| OpenLLaMA v1 | RedPajama | 1T |
| OpenLLaMA v2 | OpenLLaMA data[6] | 1T |
| TinyLlama | TinyLlama data[7] | 3T |
| Sheared-LLaMA | RedPajama | 50B |

---

[4] Please find results on LLaMA1 models in Appendix F.6 and Pythia models in Appendix F.5.

[5] OPT data contains BookCorpus (Zhu et al., 2015), Stories (Trinh & Le, 2018), CCNews (Hamborg et al., 2017), the Pile (Gao et al., 2020), and PushShift.io Reddit (Baumgartner et al., 2020).

[6] OpenLLaMA v2 is pre-trained with a mixture of RefinedWeb (Penedo et al., 2023), StarCoder (Li et al., 2023), and part of RedPajama.

[7] TinyLlama data is a mixture of SlimPajama (Shen et al., 2023) and StarCoder data.

Table 2: Sheared-LLaMA outperforms publicly available models of comparable size on downstream tasks. The shot number used is noted in parentheses, with 0-shot if not specified. Models with † use a different training data from RedPajama. Please refer to Table 1 for details.

| Model (#tokens for training) | Commonsense & Reading Comprehension | | | | | |
|---|---|---|---|---|---|---|
| | SciQ | PIQA | WinoGrande | ARC-E | ARC-C (25) | HellaSwag (10) |
| LLaMA2-7B (2T)† | 93.7 | 78.1 | 69.3 | 76.4 | 53.0 | 78.6 |
| OPT-1.3B (300B)† | 84.3 | 71.7 | **59.6** | 57.0 | 29.7 | 54.5 |
| Pythia-1.4B (300B)† | 86.4 | 70.9 | 57.4 | 60.7 | 31.2 | 53.0 |
| TinyLlama-1.1B (3T)† | **88.9** | 73.3 | 58.8 | 55.3 | 30.1 | 60.3 |
| Sheared-LLaMA-1.3B (50B) | 87.3 | **73.4** | 57.9 | **61.5** | **33.5** | **60.7** |
| OPT-2.7B (300B)† | 85.8 | 73.7 | 60.8 | 60.8 | 34.0 | 61.5 |
| Pythia-2.8B (300B)† | 88.3 | 74.0 | 59.7 | 64.4 | 36.4 | 60.8 |
| INCITE-Base-3B (800B) | 90.7 | 74.6 | 63.5 | **67.7** | 40.2 | 64.8 |
| Open-LLaMA-3B-v1 (1T) | 91.3 | 73.7 | 61.5 | 67.6 | 39.6 | 62.6 |
| Open-LLaMA-3B-v2 (1T)† | **91.8** | **76.2** | 63.5 | 66.5 | 39.0 | 67.6 |
| Sheared-LLaMA-2.7B (50B) | 90.8 | 75.8 | **64.2** | 67.0 | **41.2** | **70.8** |

| Model (#tokens for training) | Continued | | LM | World Knowledge | | Average |
|---|---|---|---|---|---|---|
| | LogiQA | BoolQ (32) | LAMBADA | NQ (32) | MMLU (5) | |
| LLaMA2-7B (2T)† | 30.7 | 82.1 | 28.8 | 73.9 | 46.6 | 64.6 |
| OPT-1.3B (300B)† | 26.9 | 57.5 | 58.0 | 6.9 | 24.7 | 48.2 |
| Pythia-1.4B (300B)† | **27.3** | 57.4 | **61.6** | 6.2 | **25.7** | 48.9 |
| TinyLlama-1.1B (3T)† | 26.3 | 60.9 | 58.8 | **12.1** | 25.5 | 50.0 |
| Sheared-LLaMA-1.3B (50B) | 26.9 | **64.0** | 61.0 | 9.6 | **25.7** | **51.0** |
| OPT-2.7B (300B)† | 26.0 | 63.4 | 63.6 | 10.1 | 25.9 | 51.4 |
| Pythia-2.8B (300B)† | 28.0 | 66.0 | 64.7 | 9.0 | 26.9 | 52.5 |
| INCITE-Base-3B (800B) | 27.7 | 65.9 | 65.3 | 14.9 | **27.0** | 54.7 |
| Open-LLaMA-3B-v1 (1T) | 28.4 | 70.0 | 65.4 | **18.6** | **27.0** | 55.1 |
| Open-LLaMA-3B-v2 (1T)† | 28.1 | 69.6 | 66.5 | 17.1 | 26.9 | 55.7 |
| Sheared-LLaMA-2.7B (50B) | **28.9** | **73.7** | **68.4** | 16.5 | 26.4 | **56.7** |

**Evaluation.** We use the `lm-evaluation-harness` package (Gao et al., 2021) to evaluate on an extensive suite of downstream tasks: (1) We follow Pythia and LLaMA2 to report the 0-shot accuracy of ARC easy (ARC-E; Clark et al., 2018), LAMBADA (Paperno et al., 2016), LogiQA (Liu et al., 2020), PIQA (Bisk et al., 2020), SciQ (Welbl et al., 2017), and WinoGrande (Sakaguchi et al., 2021). (2) We report accuracy of the tasks used by Open LLM Leaderboard (Beeching et al., 2023), including 10-shot HellaSwag (Zellers et al., 2019), 25-shot ARC Challenge (ARC-C; Clark et al., 2018), and 5-shot MMLU (Hendrycks et al., 2021). (3) We also report exact match of 32-shot Natural Questions (NQ; Kwiatkowski et al., 2019) to measure the factual knowledge in the model.

As training models to follow instructions has become a crucial application of LLMs (Ouyang et al., 2022; Taori et al., 2023), we evaluate our models on instruction tuning and fine-tune both baseline models and Sheared-LLaMA on instruction-response pairs sampled from the ShareGPT dataset.[8] Please refer to Appendix E for more details.

## 3.2 SHEARED-LLAMA OUTPERFORMS LMS OF EQUIVALENT SIZES

We demonstrate that Sheared-LLaMA outperforms existing LLMs of similar sizes on both standard LLM benchmarks and instruction tuning, while using only a fraction of the compute budget required to train those models from scratch.

**Downstream tasks.** Table 2 presents the zero-shot and few-shot downstream task performance of Sheared-LLaMA and similarly-sized pre-trained models. Even with a limited budget of approximately 50B tokens for pruning and continued pre-training, Sheared-LLaMA models outperform existing models pre-trained on significantly larger compute. Sheared-LLaMA-1.3B outperforms OPT-1.3B, Pythia-1.4B (pre-trained with 300B tokens), and TinyLlama-1.1B (pre-trained

---

[8]https://sharegpt.com/

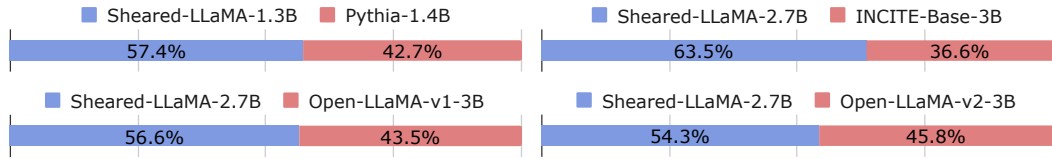

Figure 3: Sheared-LLaMAs outperform Pythia-1.4B, INCITE-Base-3B, OpenLLaMA-3B-v1 and OpenLLaMA-3B-v2 in instruction tuning.

on 3T tokens). Sheared-LLaMA-2.7B outperforms INCITE-Base-3B (pre-trained on 800B Red-Pajama tokens), OpenLLaMA-3B-v1 (pre-trained on 1T RedPajama tokens), and OpenLLaMA-3B-v2 (trained on 1T tokens from RedPajama, RefinedWeb, and StarCoder). The most noteworthy result is that Sheared-LLaMA-1.3B outperforms TinyLlama-1.1B, despite TinyLlama-1.1B being pre-trained on 3T tokens—more than the total data used for pre-training LLaMA2-7B and our pruning process combined. This demonstrates that structured pruning is a more *sample-efficient* approach for training smaller-scale LLMs.

**Instruction tuning.** As shown Figure 3, instruction-tuned Sheared-LLaMA achieves higher win rates compared to all the other pre-trained models at a comparable scale. This demonstrates that our 2.7B model can serve as a strong foundation for instruction tuning and has the capacity to generate long, coherent and informative responses (See examples in Appendix E).

# 4 ANALYSIS

## 4.1 EFFECTIVENESS OF DYNAMIC BATCH LOADING

We analyze the effectiveness of dynamic batch loading by examining its impact on three aspects: (1) the final LM loss across domains, (2) the data usage of each domain throughout training, (3) the downstream task performance. All results in this section are based on Sheared-LLaMA-1.3B.[9]

**Loss differences across domains.** Dynamic batch loading aims to balance the rate of loss reduction across domains, ensuring that the losses reach the reference value at roughly the same time. Figure 4 shows the difference between our model's loss (with both original and dynamic batch loading) and the reference loss, estimated by fitting a scaling function to a hypothetical 1.3B parameter LLaMA2 model. The original batch loading results in widely varying loss differences across domains; for example, the GitHub loss decreases below the reference value, while the C4 loss lags behind. Dynamic batch loading, however, reduces losses evenly and leads to very similar loss differences across domains, suggesting more efficient data use.

**Data usage.** Table 3 compares the data proportion of RedPajama and the data usage of our dynamic loading approach (Figure 6 illustrates how the domain weights change during training). It shows that dynamic batch loading loads more data from the Book and C4 subsets, indicating that these domains are more challenging for a pruned model to recover.

Table 3: Domain data usage with dynamic batch loading compared to the original proportions.

|                       | CC     | GitHub | Book  | StackExchange | Wiki  | ArXiv | C4     |
|-----------------------|--------|--------|-------|---------------|-------|-------|--------|
| RedPajama (Original)  | 67.0%  | 4.5%   | 4.5%  | 2.0%          | 4.5%  | 2.5%  | 15.0%  |
| Dynamic Batch Loading | 36.1%  | 0.8%   | 9.1%  | 1.0%          | 3.1%  | 0.7%  | 49.2%  |

**Downstream performance.** As shown in Figure 5, pruned models trained with dynamic batch loading achieve better downstream performance than when trained on the original RedPajama distribution. This suggests that the more balanced loss reduction from dynamic batch loading transfers to improved downstream capabilities.

---

[9]We also experiment with a heuristic approach to exclude the easy domains from pruning, but find that the loss disparaty issue persists after continued pre-training. Please refer to Appendix F.8 for mode details.

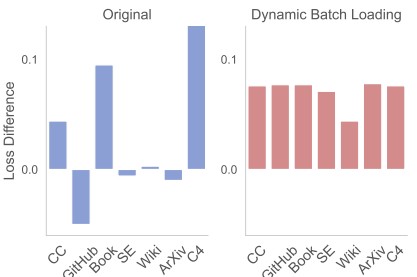 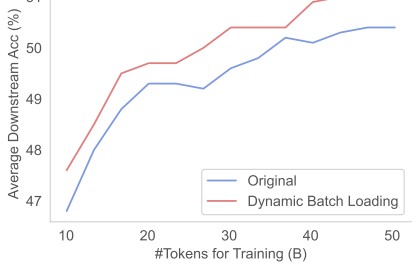

Figure 4: Loss difference between the pruned model (1.3B) and estimated reference loss, with original vs. dynamic batch loading.

Figure 5: Downstream task performance of Sheared-LLaMA-1.3B with original data proportion and dynamic batch loading.

## 4.2 Comparison to Other Pruning Approaches

We compare our LLM-shearing to other pruning approaches on validation perplexity, a strong indicator of overall model capabilities (Xia et al., 2023).

**Targeted pruned models have a higher inference speed.** Previous works like CoFiPruning (Xia et al., 2022) produce structed pruned models, but these models often have non-uniform layer configurations (e.g., different numbers of heads across layers). Such non-uniformity across layers introduces training and inference overhead due to irregularities in model architectures. We experiment with both CoFiPruning and targeted structured pruning, with a 0.4B pruning budget with the Red-Pajama data proportion for a fair comparison. Table 4 shows that our targeted pruned models have a higher inference speed compared to the non-uniformly pruned CoFiPruning model at the same sparsity, despite having a slightly higher perplexity. Targeted structured pruning needs about 0.5B more tokens in continued pre-training to match CoFiPruning's perplexity. However, this one-time extra compute during training is justified, as it results in a more efficient model architecture that is essential for real-world applications and effective practical use. Please find more details on inference speed of different pruning methods in Appendix F.9.

Table 4: Validation perplexity and generation speed during inference (tokens/second) of targeted structured pruning with a uniform layer configuration, and CoFiPruning, with a non-uniform layer configuration. Inference speed is measured on a Nvidia A100 (80G) GPU, on a singal instance generating up to 512 tokens.

|      | Layer Config      | PPL ↓ | Speed ↑ |      | Layer Config      | PPL ↓ | Speed ↑ |
|------|-------------------|-------|---------|------|-------------------|-------|---------|
| 1.3B | CoFiPruning       | 9.1   | 51      | 2.7B | CoFiPruning       | 7.0   | 37      |
|      | Targeted pruning  | 10.3  | 58      |      | Targeted pruning  | 7.7   | 43      |

**Comparison to LLM-Pruner (Ma et al., 2023).** We compare targeted structured pruning to LLM-Pruner, a recent work in structured pruning, in Appendix F.2. We demonstrate that, given the same compute budget, sparsity level, and training data distribution, our pruned models achieve lower perplexity, have a more optimized architecture, and faster inference speed.

## 4.3 Additional Analysis

**Budget allocation for pruning and continued pre-training.** Intuitively, allocating more compute to the pruning stage helps identify better subnetwork structures. We explore distributing data across pruning and continued pre-training stages differently, within a fixed budget of 5B tokens. Table 5 shows that when controlling the total amount of tokens, increasing the pruning budget consistently improves perplexity. However, since pruning is more expensive than continued pre-training, we decide to allocate 0.4B tokens to pruning. Please refer to Appendix C for details on training throughputs.

Table 5: Data budget allocation to pruning and continued pre-training (CT) and corresponding perplexity.

| # Tokens |      | PPL      |      |
|----------|------|----------|------|
| Pruning  | CT   | Pruning  | CT   |
| 0.2B     | 4.6B | 12.99    | 7.46 |
| 0.4B     | 4.4B | 10.29    | 7.32 |
| 0.8B     | 4.0B | 9.01     | 7.23 |
| 1.6B     | 3.2B | 8.04     | 7.08 |

**More analysis.** We provide further analysis in the appendix: (1) Sheared-LLaMA evaluation on math and coding (Appendix F.3), (2) Pythia model pruning (Appendix F.5), and (3) impact of excluding easy domains during pruning (Appendix F.8).

## 5 RELATED WORK

**Pruning.** Structured pruning has been extensively studied as a model compression technique in computer vision and natural language processing, where task-specific models like classification ones are often overparameterized and can be pruned significantly with minimal impact on performance (Han et al., 2016; Wen et al., 2016; Liu et al., 2017; Luo et al., 2017; Cai et al., 2019; Deng et al., 2020; Hou et al., 2020; Wang et al., 2020; Lagunas et al., 2021; Xia et al., 2022; Kurtic et al., 2023). Unstructured pruning (Frankle & Carbin, 2018; Li et al., 2020; Chen et al., 2020; Sanh et al., 2020) prunes individual neurons instead of structured blocks. Though unstructured pruning usually achieve higher compression rates, they are not practical for model speedup.

In the era of LLMs, the prevalent NLP pipeline has shifted from task-specific models to general-purpose LMs, which leaves little room for redundancy. Both unstructured pruning, semi-structured pruning (Frantar & Alistarh, 2023; Sun et al., 2023), and structured pruning (Ma et al., 2023) lead to significant performance drops on LLM even at a modest sparsity. Noticeably, all previous works fix the original models or tune them minimally. We see pruning as an initialization and consider it necessary to expend substantial compute to continually pre-training the model to recover performance.

**Efficient pre-training approaches.** As orthogonal to our pruning approach, There is an extensive body of work on improving efficiency of training LLMs. For example, quantization reduces the numeric precision of model weights and activations and speeds up training and inference (Dettmers et al., 2022; 2023; Xiao et al., 2023). Knowledge distillation (Hinton et al., 2015; Sanh et al., 2019; Jiao et al., 2020; Sun et al., 2020), which trains a smaller model on a larger model's prediction, is shown to be effective for task-specific models (Xia et al., 2022). For pre-training LLMs, though distilling from a teacher model is shown to improve the quality of student models given the same number of training steps (Rae et al., 2021; Blakeney et al., 2022), it is less cost-effective than pruning and continued training due to the exceeding inference cost incured by the teacher model (Jha et al., 2023). More methods have been introduced to enhance the efficiency of training LMs, such as dynamic architectures (Gong et al., 2019; Zhang & He, 2020) and efficient optimizers (Chen et al., 2023; Liu et al., 2023). However, as indicated by (Kaddour et al., 2023; Bartoldson et al., 2023), the promised gains in training efficiency may not be consistently realized.

There are also data-based approaches to enhance training efficiency. Eliminating duplicated data is found to be effective (Lee et al., 2021). Various batch selection techniques propose to prioritize data based on criteria such as higher losses (Jiang et al., 2019) or a greater reducible loss (Mindermann et al., 2022). Xie et al. (2023) propose to optimize data mixtures by training a proxy model to estimate the optimal data weight of each domain.

## 6 DISCUSSION

**Limitation and future work.** First, The method heavily depends on the availability of open-source pre-training datasets and models. If a specific domain is not covered in the pre-training data, the method may not recover performance well on that domain. Second, Due to computational constraints, we only experimented with a 7B parameter model as the source model. However, our method is highly generalizable and can be scaled up to larger models in future research.

**Conclusion.** This work proposes structured pruning as an efficient method for creating competitive smaller-scale LLMs. Our two-stage approach combines targeted structured pruning and continued pre-training (continued pre-training), and we introduce *dynamic batch loading* to improve pre-training data efficiency. We train a series of competitive Sheared-LLaMA models using a fraction of the compute required for standard pre-training. Our results show a promising path to producing low-cost, small LLMs when strong large-scale models are available. As more capable LLMs and larger pre-training datasets emerge, our method can easily extend to these advances to create even better small models.

## ACKNOWLEDGEMENTS

We express our gratitude to Sadhika Malladi, Tanya Goyal, Ofir Press, Adithya Bhaskar, and the Princeton NLP group for reviewing the paper and providing helpful feedback. We also thank the engineering team at MosaicML for their invaluable assistance with implementation specifics using the Composer package. Mengzhou Xia is supported by a Bloomberg Data Science Ph.D. Fellowship, and Tianyu Gao is supported by an IBM PhD Fellowship. This research is also supported by Microsoft Azure credits through the "Accelerate Foundation Models Academic Research" Initiative.

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

CONTENTS

## A   A DETAILED EXPOSITION OF PARAMATERIZING PRUNING MASKS

The key idea behind the pruning algorithm is to apply masks to the model parameters. After learning a binary mask, it is equivalent to removing the corresponding parameters. The mask is parameterized using a hard concrete distribution introduced in Louizos et al. (2018). Given a masking variable $z$ parameterized by $\alpha$, the hard concrete distribution is defined as follows:

$$u = \mathcal{U}(0, 1)$$
$$s = \text{Sigmoid}\left(\frac{1}{\beta}\left(\log\frac{u}{1-u} + \log\alpha\right)\right)$$
$$\bar{s} = s(\zeta - \gamma) + \gamma$$
$$z = \min(1, \max(0, \bar{s}))$$

where $\mathcal{U}$ is the uniform distribution, $\beta$ is a temperature parameter, $s$ is a relaxed binary mask that conforms to the hard concrete distribution, and $\zeta$ and $\gamma$ are the bounds of the hard concrete distribution. The hard concrete distribution serves as a continuous relaxation of the binary mask, allowing the model to learn the binary mask in a continuous manner during training. The effectiveness of this trick in learning sparse structures in neural networks has been demonstrated in previous studies (Wang et al., 2020; Xia et al., 2022). In our experiments, we set $\beta = 0.83$, $\zeta = 1.1$, and $\gamma = -0.1$.

To enforce the sparsity constraint, the masks are trained alongside with Lagrange multipliers $\lambda$, as defined in Equation (1). After pruning, the parameters corresponding to the learned masks are removed to ensure that the resulting model shape matches the target model. In practical implementations, we set a threshold to binarize the masks. Due to the adoption of the hard concrete distribution, the masks typically converge to binary values that match the target model shape in most cases, thereby avoiding any inconsistencies. However, in rare instances where the masks do not converge to exactly 0 or 1, the masking variables need to be absorbed into the resulting model parameters.

As discussed in Section 2, we apply masks to heads, intermediate dimensions, layers and hidden dimensions. For heads, we simply multiply the head output by the mask. For intermediate dimensions, we apply the mask to the intermediate output. For layers, we apply the mask to the layer output. For hidden dimensions, we apply the mask to both the head and mlp output. Applying the mask to outputs is equivalent to removing the corresponding parameters. Please refer to composer_llama.py for more details.

## B   REFERENCE LOSS PREDICTED BY SCALING LAWS

The scaling law of language modeling is a function of model size $N$ and dataset size $D$:

$$L(N, D) = E + \frac{A}{N^\alpha} + \frac{B}{D^\beta}$$

where $E$ captures the loss for the true language distribution in an ideal generation process, and $A, \alpha, B, \beta$ are scaling factors related to model scale or data size. Models in the same model family are usually trained with the same amount of tokens on the same data distribution. In this case, we need a minimum of three models to estimate the constant $E + \frac{B}{D^\beta}$, $A$ and $\alpha$. If the models are trained with different amount of tokens, we can estimate $E, A, \alpha, B, \beta$ with a minimal of 5 models. Note that we will estimate the scaling factors for each domain seperately.

LLAMA2 models have been trained on the same 2T tokens (Touvron et al., 2023b). We take the LLAMA2-7B, LLAMA2-13B, and LLAMA2-70B checkpoints, evaluate them on each domain's validation set, and fit the scaling factors with the corresponding loss. Given the limited data points for estimating the scaling law constant, the projected loss of a hypothetical LLaMA-2.7B model may be biased compared to the true value. Table 6 presents the predicted loss. The evaluation process takes less than 4 A100 GPU hours to complete.

Table 6: Estimated reference loss of hypothetical LLaMA2-1.3B and LLaMA2-2.7B models.

| | CC | GitHub | Book | StackExchange | Wiki | ArXiv | C4 |
|---|---|---|---|---|---|---|---|
| LLaMA2-1.3B | 1.964 | 0.746 | 2.139 | 1.612 | 1.759 | 1.445 | 2.125 |
| LLaMA2-2.7B | 1.871 | 0.688 | 2.033 | 1.535 | 1.630 | 1.356 | 2.033 |

## C  TRAINING DETAILS

We present the hyperparameters used in our experiments in Appendix C. We use fully sharded data parallel (Zhao et al., 2023) to train our models in parallel. We use FlashAttention V1 (Dao et al., 2022) to speed up training. We use a cosine learning rate scheduler and decay the learning rate to a minimum of 10% of the peak value. We conduct some preliminary experiment to determine the peak learning rate for learning the masking variables and Lagrange multiplers, and we find that a learning rate of 1.0 works well for pruning. We do not tune any other hyper-parameters. The throughput is dependent on the implementations and we believe that our throughput can be further improved by adopting more advanced recent optimizations such as FlashAttention V2 (Dao et al., 2022) and a more recent version of Composer (MosaicML, 2021).

Table 7: Training hyper-parameters and throughput.

| | Pruning | Contined Pre-training |
|---|---|---|
| Training budget | 0.4B | 50B |
| Learning rate of $z, \phi, \lambda$ | 1.0 | - |
| Learning Rate of $\theta$ | 0.0001 | 0.0001 |
| LR warmup ratio | 10% | 3% |
| Batch size (tokens) | 131K | 1M |
| Evaluation interval $m$ (steps) | 50 | 400 |
| Steps | 3, 200 | 51, 200 |
| # GPUs | 8 | 16 |
| Throughput (tokens/s) | 15K | 145K (1.3B) / 77K (2.7B) |

## D  MODEL CONFIGURATIONS

In this section, we provide the model configurations for both our Sheared-LLaMA models and the baseline models, as illustrated in Table 8. Our design closely adheres to the architecture of Pythia-1.4B and INCITE-Base-3B, albeit with some nuanced distinctions. A noteworthy difference is found in the intermediate size of Sheared-LLaMA, which is a consequence of its lineage from LLaMA2-7B. Notably, LLaMA2-7B employs a GLU variant (Shazeer, 2020) within its feed-forward layer, comprising a gate matrix, an upward-projection matrix, and a downward-projection matrix. In contrast, other models employ the conventional double-matrix feed-forward layer structure. Furthermore, we acknowledge that the shearing algorithm will have to inherit the head dimension of the source model. Instead of explicitly specifying the number of heads based on existing language models, we set the target number of heads to be the target hidden dimension divided by the head dimension of the source model.

## E  INSTRUCTION TUNING

We evaluate our models on instruction tuning and fine-tune both Sheared-LLaMA and baseline models on 10,000 instruction-response pairs sampled from the ShareGPT dataset[10]. For evaluation, we sample another 1,000 instructions from ShareGPT, generate responses from our fine-tuned models and other baseline models, and use GPT-4 as an evaluator to compare the two responses (Dubois et al., 2023). We report the win rate of our model compared to the baseline model.

During instruction tuning training, the instruction is prepended with "You are a helpful assistant. Write a response that appropriately completes the request.". For evaluating the instruction tuning

---

[10] https://sharegpt.com. We only use the first round in the multi-turn chat history.

Table 8: Model configurations of our Sheared-LLaMA and baseline models.

| Model | #Param | #Layers | Hidden | Intermediate | #Heads | Head Dim |
|---|---|---|---|---|---|---|
| OPT-1.3B | 1.3B | 24 | 2048 | 8192 | 32 | 64 |
| Pythia-1.4B | 1.4B | 24 | 2048 | 8192 | 16 | 128 |
| TinyLlama-1.1B | 1.1B | 22 | 2048 | 5632 | 32 | 64 |
| Sheared-LLaMA-1.3B | 1.3B | 24 | 2048 | 5504 | 16 | 128 |
| OPT-2.7B | 2.7B | 32 | 2560 | 10240 | 32 | 80 |
| Pythia-2.8B | 2.8B | 32 | 2560 | 10240 | 32 | 80 |
| INCITE-Base-3B | 2.8B | 32 | 2560 | 10240 | 32 | 80 |
| OpenLLaMA-3B | 2.7B | 26 | 3200 | 8640 | 32 | 100 |
| Sheared-LLaMA-2.7B | 2.7B | 32 | 2560 | 6912 | 20 | 128 |
| LLaMA2-7B | 6.7B | 32 | 4096 | 11008 | 32 | 128 |

generations, Wang et al. (2023a) observes using GPT models as a judge could change its preference when swapping the presentation order of the two outputs. Therefore, we compare each output pair twice by swapping the presentation order of the two outputs and finally report the average win-rate of the two rounds to eliminate the position bias.

We randomly select an output generated by Sheared-LLaMA-1.3B and Sheared-LLaMA-2.7B in response to a given instruction, and present the generations in Table 10. Our findings demonstrate that, after instruction tuning, Sheared-LLaMA-2.7B consistently produces long, coherent, and informative outputs in response to the instruction.

Table 9: Training hyper-parameters for instruction tuning.

| | Instruction Tuning |
|---|---|
| Learning Rate of $\theta$ | $5e - 5$ |
| LR warmup ratio | $3\%$ |
| Batch size (tokens) | 128 |
| # GPUs | 8 |

## F  ADDITIONAL EXPERIMENT RESULTS

### F.1  DATA USAGE IN CONTINUED PRE-TRAINING

Figure 6 illustrates the evolution of domain weights throughout the training process and the final cumulative data usage for each domain. The trajectory reveals that the domain weights stabilize after approximately 30% of the training. In contrast to other domains, Wikipedia exhibits an anomalous spike early in the training process. The remaining domains show a steady, monotonic change in data loading over time, as expected.

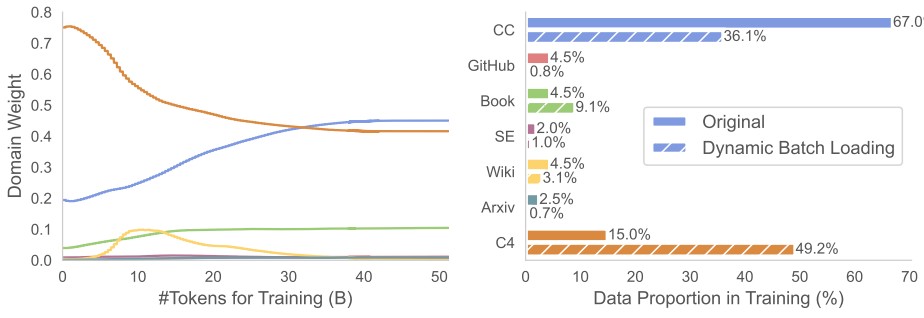

Figure 6: Left: Data weight of each batch during the continued pre-training stage. Right: Cumulative data usage for each domain.

Table 10: An example output from Sheared-LLaMA-1.3B and Sheared-LLaMA-2.7B. It shows that Sheared-LLaMA can generate long, coherent, and informative responses after instruction tuning.

```
Prompt:  Act as semiconductor industry analyst

Sheared-LLaMA-1.3B: As a semiconductor industry analyst, I would be
responsible for conducting market research and analysis to help
companies in the industry understand and respond to market trends,
customer preferences, and competitive dynamics.  This would include
analyzing market data such as sales, market share, and industry trends,
as well as conducting interviews with industry leaders, executives, and
customers to gain insights into customer needs and preferences.  I would
also be responsible for developing and implementing marketing strategies
to promote products and services, and for identifying and pursuing new
business opportunities.

Sheared-LLaMA-2.7B: As a semiconductor industry analyst, my primary
responsibility is to provide insightful and accurate analysis on the
global semiconductor market.  This includes evaluating the performance
of individual companies, industries, and trends within the market.

Here are some key areas where I would focus my analysis:

1.  Market size and growth:  I would track the size of the global
semiconductor market and analyze the growth prospects for each segment,
such as memory, logic, and sensors.
2.  Competitive landscape:  I would assess the competitive landscape,
including the top players, their market share, and strategic moves.
3.  Technology trends:  I would monitor the development of new
technologies, such as 5G, AI, and IoT, and their impact on the
semiconductor industry.
4.  Industry trends:  I would track industry trends, such as the shift
towards sustainable technologies, and evaluate their impact on the
market.
5.  Market forecasts:  I would provide forecasts for the semiconductor
market, including revenue and unit shipment growth, based on a range of
scenarios and assumptions.

In addition to these areas, I would also keep track of industry news and
developments, and provide insights on their impact on the market.
```

## F.2   COMPARISON TO LLM-PRUNER

To ensure a fair comparison with the LLM-Pruner approach, we match the parameters (excluding embeddings) to be roughly the same as our final model (1.23B), as embedding sizes do not affect inference speed. We continue pre-training the pruned models derived from both LLM-Pruner and our proposed targeted structured pruning. The total number of tokens for pruning and continued pre-training is controlled to be the same, and data from the RedPajama dataset is used directly without applying dynamic batch loading. We demonstrate that our proposed targeted structured pruning is a better approach compared to LLM-Pruner from three aspects: the loss trajectory, the model architecture, and the inference speed.

In terms of loss trajectory, Figure 7 shows that our proposed targeted structured pruning achieves a lower loss than LLM-Pruner when consuming the same amount of data.

Table 11 compares the model configurations for an LLM-Pruner pruned model and our pruned model. The LLM-Pruner model has an unconventional architecture where the intermediate size is smaller than the hidden size, largely due to the algorithm's inability to prune the hidden dimension and layers, revealing a limitation of LLM-Pruner.

In terms of training throughput and inference speed, we find Sheared-LLaMA structures run more efficiently than LLM-Pruner models. We performed an inference speed analysis comparing LLM-pruner and Sheared-LLaMA's model architectures using a single A100 GPU to generate up to 2048 tokens. As shown in Table 12, our pruned model architecture is significantly more efficient than

LLM-Pruner at inference time. Additionally, LLM-Pruner's model architecture introduces substantial overhead during continued pretraining (Measured with 16 A100 80GB GPUs.), with a training throughput of around 60% of Sheared-LLaMA's. Overall, our Sheared-LLaMA architecture enables higher throughput for both inference and continued training compared to LLM-Pruner.

In summary, we have demonstrated that at the same parameter scale, our pruning method produces a model that has a lower perplexity (loss), a more reasonable final model architecture, and a faster inference speed. We have effectively shown our targeted structured pruning algorithm to be more effective for large-scale LLM pruning compared to LLM-Pruner.

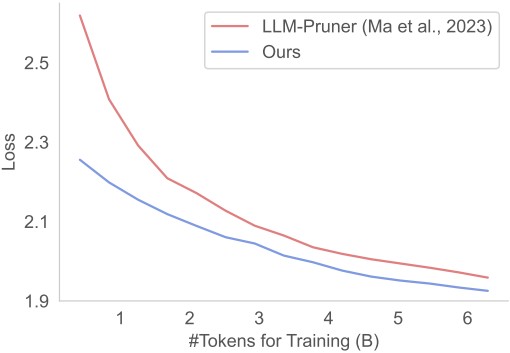

Figure 7: The loss of LLM-Pruner and Sheared-LLaMA during the continued pre-training stage. Note that we exclude dynamic batch loading and use the same data distribution for training both models for a fair comparison.

Table 11: Model structure of Pythia-1.4B, LLM-pruner (1.6B), and Ours (1.3B).

| Layers | Heads | Head size | Intermediate size | Hidden size | Params | |
|---|---|---|---|---|---|---|
| Pythia-1.4B | 24 | 16 | 128 | 8192 | 2048 | 1.4B |
| LLM-pruner (1.6B) | 32 | 7 | 128 | 2201 | 4096 | 1.6B |
| Sheared-LLaMA (1.3B) | 24 | 16 | 128 | 5504 | 2048 | 1.3B |

Table 12: Training throughput and generation speed of LLM-pruner (1.6B) and Sheared-LLaMA (1.3B). With a similar parameter count, our pruned model structure has a lower perplexity when fine-tuned with the same amount of tokens (around 6B tokens). Yet our pruned model architectures are way more efficient for both training and inference.

| | Generation Speed | Training Throughput | PPL |
|---|---|---|---|
| LLM-Pruner | 43 tokens/s | 83K tokens/s | 7.09 |
| Sheared-LLaMA | 58 tokens/s | 139K tokens/s | 6.85 |

### F.3 Coding and Math Reasoning

We examine the math and coding abilities of our pruned models compared to other language models. We find that the math ability of existing 3B parameter models, including Sheared-LLaMA, is still far below that of larger models. We also find that Sheared-LLaMA's coding ability lags behind models known to be trained on more code data, like Pythia-1.4B and Open-LLaMA-3B-v2. Sheared-LLaMA's coding ability likely comes from the original LLaMA2 model, speculated to have used more code data, and the minimal code data used in our pruning experiments.

### F.4 Scaling Reference vs. Source Reference

Figure 8 This section compares the performance of Sheared-LLaMA when trained with the scaling reference and the source reference in dynamic batch loading. The scaling reference uses the predicted loss from the scaling law as the reference loss, while the source reference uses the loss of the

Table 13: Evaluation results on GSM8K and HumanEval and training percentage and tokens in ArXiv and GitHub.

| Models | GSM8K (8) EM | HumanEval Pass@1 | HumanEval Pass@5 | ArXiv Percentage | Github Percentage | ArXiv Tokens | GitHub Tokens |
|---|---|---|---|---|---|---|---|
| LLaMA2-7B | 13.7 | 12.8 | 23.8 | - | - | - | - |
| OPT-2.7B | 0.1 | 0.0 | 0.0 | - | - | - | - |
| Pythia-2.8B | 1.7 | 5.1 | 14.6 | 9.0% | 7.6% | 26.9 | 22.8 |
| INCITE-Base-3B | 1.8 | 4.3 | 4.9 | 2% | 4.5% | 16.0 | 36.0 |
| Open-LLaMA-3B-v1 | 2.5 | 0.0 | 1.2 | 2% | 4.5% | 20.0 | 45.0 |
| Open-LLaMA-3B-v2 | 2.7 | 10.4 | 20.1 | - | - | - | - |
| Sheared-LLaMA-2.7B (Source) | 2.7 | 3.7 | 5.5 | 0.7% | 0.4% | 0.3 | 0.2 |
| Sheared-LLaMA-2.7B (Scaling) | 2.4 | 4.9 | 9.2 | 1.0% | 0.8% | 0.5 | 0.4 |

source model as the reference loss. Although both methods efficiently train the model, the scaling reference consistently achieves slightly better downstream performance.

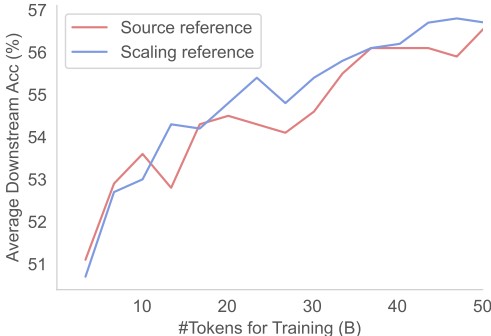

Figure 8: Average downstream peformance of Sheared-LLaMA-1.3B with the scaling reference and the source reference.

## F.5 PRUNING PYTHIA MODELS

During the initial development of the approach, we experimented with a smaller-scale model on Pythia (Biderman et al., 2023), a series of open-source models with open-source training data across scales from 70M to 13B. We took the Pythia-440M model, pruned it down to 160M parameters, and continued pre-training it using Pythia models' training data Gao et al. (2020). Specifically, we used 0.4B tokens for pruning and 33B tokens (32,000 steps) for continued pre-training of the pruned model. Table 14 shows that the pruned model achieves a lower perplexity than the original model, and continued pre-training further improves performance. Notably, with minimal compute consumption (10B tokens), pruning a Pythia-410M model reaches roughly the same performance as pretraining Pythia-160M from scratch. Adding more tokens further enhances the performance.

Table 14: Zero-shot performance of Pythia-160M and Sheared-Pythia.

| | Training Tokens | Performance |
|---|---|---|
| Pythia-160M | 300B | 43.56 |
| Sheared-Pythia | (300B) + 10B | 43.51 |
| Sheared-Pythia | (300B) + 33B | **45.78** |

Additionally, we compared Sheared-Pythia-160M against keeping pre-training the Pythia-160M model with the same amount of tokens. From Figure 9, we can see that continuing pre-training Pythia-160M starts off performing better, however, the Sheared-Pythia-160M learns faster and eventually exceeds the performance of continuing pretraining on Pythia-160M. These are some very preliminary results we see in this particular setting.

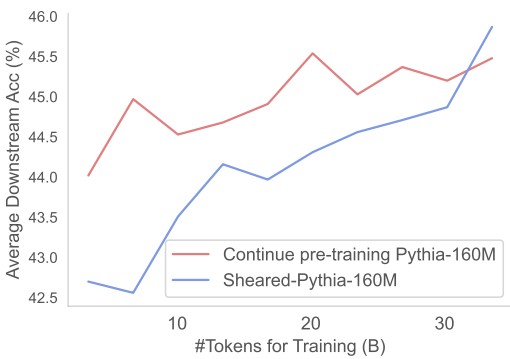

Figure 9: The downstream performance of continued pre-training Pythia-160M and Sheared-Pythia-160M. Sheared-Pythia-160M eventually outperforms the performance of continued pre-training Pythia-160M.

We think that the benefit of pruning a larger model will be even more significant, based on the conclusions from a previous work (Li et al., 2020) showing that pruning larger than compress leads to better performance as the larger models are easier to optimize. However, we'd like to defer more detailed analysis to future work.

## F.6 PRUNING FROM LLaMA1 vs LLaMA2

This section compares the performance of pruning from LLaMA1 and LLaMA2. Both models demonstrate strong downstream task performance, although pruning from LLaMA2 unsurprisingly yields a consistent advantage. However, it is worth noting that the performance difference between the two is not very large.

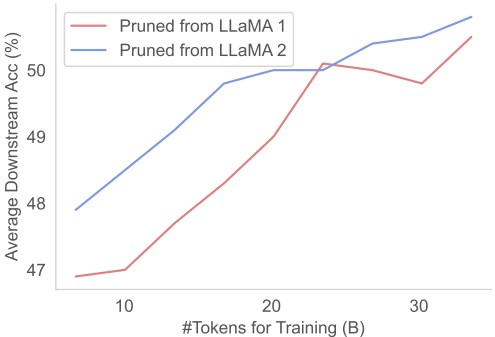

Figure 10: A comparison between pruning from LLaMA1 and LLaMA2 with dynamic loading for 1.3B.

## F.7 COMPARISON TO FURTHER CONTINUAL PRE-TRAINING INCITE-BASE-3B

We examine if pruning produces a better initialization for continued pre-training than an existing LLM of equivalent size by comparing the performance of a continually pre-trained INCITE-Base-3B model and Sheared-LLaMA-2.7B. We present the loss curves in Figure 11 and the downstream performance in Figure 12. INCITE-Base-3B model starts with higher task accuracy but plateaus after training, while Sheared-LLaMA rapidly improves and surpasses the INCITE-Base-3B model, suggesting that pruned models from a strong base model serve as a better initialization.[11]

---

[11]In cases where the existing small model is competitive compared to the pruning source model, the small model may offer a better starting point than a pruned model. Intuitively, the larger the discrepancy in performance between the source model and the small model, the more advantages the pruned model has.

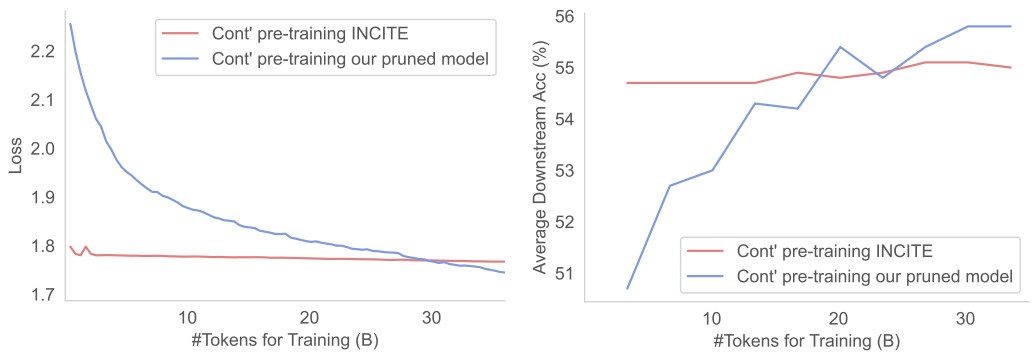

Figure 11: The loss of continued pre-training INCITE-3B and our pruned LLaMA model. Both models have around 2.7B parameters.

Figure 12: Average downstream performance of continuing pre-training Sheared-LLaMA vs INCITE-Base-3B.

We used a learning rate $1e - 5$ for continued pre-training INCITE-Base-3B, along with a scheduler to warm up the learning rate to $1e - 5$ in the first $3\%$ of the training steps, and follows a cosine decay schedule. In hindsight, how we continued pre-training the INCITE-Base-3B model may not be optimal according to recent research (Gupta et al., 2023).

## F.8 EXCLUDING EASY DOMAINS DURING PRUNING

During the development of this project, we explored an easy and intuitive idea to address the imbalanced loss decreasing rate during pruning and continued pre-training. Specifically, we excluded GitHub, StackExchange, and ArXiv data during pruning since these three domains' losses decrease the fastest. We pruned LLaMA1-13B down to 7B using a composite dataset of C4, CC, Wiki, and Books, with a heuristically constructed proportion of $40\%, 40\%, 10\%, 10\%$, respectively. We then continued pre-training the pruned model on the RedPajama dataset, which includes the excluded domains during pruning.

The results showed that the perplexity difference was more even across domains when pruning without using data from these three domains. However, after continued pre-training with all data from the seven domains in the RedPajama dataset, the loss disparity grew, with the GitHub difference being much smaller than domains like C4. These results demonstrate that simply excluding the domains that are easy to recover during the pruning stage does not inherently resolve the imbalance of loss difference across domains.

This set of experiments motivated us to develop dynamic batch loading as a more effective and principled approach to address the domain-specific loss disparities that arise during pruning and continued pre-training.

Table 15: Pruning LLaMA1-13B with a composite of $40\%$ of CC, $40\%$ of C4, $10\%$ of Books and $10\%$ of Wikipedia to a 7B model. We present the domain loss of the source model (LLaMA1-13B), the loss of the pruned model and the loss after continued pre-training of the pruned model. The loss differentce from the target model (LLaMA1-7B) is more balanced after pruning, but more disparate after continued pre-training with all the domains.

|  | CC | GitHub | Book | StackExchange | Wikipedia | ArXiv | C4 |
|---|---|---|---|---|---|---|---|
| LLaMA1-13B | 1.7585 | 0.6673 | 1.9499 | 1.4207 | 1.4331 | 1.3855 | 1.8619 |
| LLaMA1-7B | 1.8366 | 0.7108 | 2.0322 | 1.5112 | 1.5291 | 1.4340 | 1.9331 |
| Pruned model (w/o three domains) | 2.1849 | 1.0971 | 2.3726 | 1.9080 | 2.1151 | 1.7542 | 2.3187 |
| diff from LLaMA1-7B | 0.3483 | 0.3863 | 0.3404 | 0.3968 | 0.5860 | 0.3202 | 0.3857 |
| Continued Pretraining (w RP) | 1.8344 | 0.6325 | 2.0984 | 1.4542 | 1.4549 | 1.4460 | 2.0395 |
| diff from LLaMA1-7B | -0.0022 | -0.0783 | 0.0661 | -0.0570 | -0.0743 | 0.0120 | 0.1064 |

## F.9 INFERENCE SPEED ANALYSIS

In this section, we analyze the inference speed of different pruning approaches, including the following models:

- The source model, i.e., LLaMA2-7B.
- Sheared-LLaMA-1.3B and Sheared-LLaMA-2.7B.
- Wanda pruning (Sun et al., 2023) to prune LLMs into a semi-structured 2:4 and 4:8 sparsity pattern in one-shot.
- LLM-Pruner (Ma et al., 2023), which produces a model with the same number of non-embedding parameters as Sheared-LLaMA.

We use an A100 GPU to test the generation speed (tokens/second) of all these pruned models. We generate up to 2048 tokens with a batch size of 1. We present the results in Table 16. Sheared-LLaMA's speed is better than that of LLM-Pruner, largely due to the more optimized resulting architecture. As shown in Table 11, LLM-pruner produces a model structure with a smaller intermediate size than the hidden size, which goes against the transformer designs where the intermediate size is at least 3-4 times the hidden size.

Wanda-type semi-structured pruning also achieves inference speedup compared to the source model. However, it is not as fast as small dense models and is less flexible because inference speedup is only feasible when the sparsity is at $50\%$.

Table 16: Inference speed (tokens/s) of different pruning approaches.

| Model | Throughput | |
|---|---|---|
| | **7B** | |
| LLaMA-7B | 37 | |
| | **1.3B** | **2.7B** |
| LLM Pruner | 41 | 40 |
| Sheared-LLaMA | 62 | 47 |
| | **50% sparsity** | |
| Wanda (2:4) | - | 42 |
| Wanda (4:8) | - | 42 |

## G FREQUENTLY ASKED QUESTIONS

In this section, we provide answers to frequently asked questions about our work.

▷ **Is it fair to say that Sheared-LLaMA models can be produced using only 50B tokens, even though the source model (LLaMA2) was trained on 2T tokens?**

At the time of our paper submission, there were no models sufficiently trained for 2T tokens at the 1.3B and 2.7B scale to allow for a fair comparison. However, the recently released TinyLlama-1.1B models, trained on 3T tokens, provide a suitable point of reference. We observe that the performance of TinyLlama-1.1B is comparable to Sheared-LLaMA-1.3B on downstream benchmarks when used as base models, and a similar observation can be found in Wang et al. (2023b). Considering that TinyLlama-1.1B is trained with 3T tokens, which exceeds the total amount of pre-training and pruning used by Sheared-LLaMA-1.3B (2T for pre-training the source model, and 50.4B for pruning and continued training), we regard this as strong evidence suggesting that pruning might be an intrinsically more efficient and effective approach to training moderate-sized LMs.

▷ **How is dynamic batch loading different from Doremi (Xie et al., 2023)?**

Dynamic batch loading and Doremi share the same principle, which adjusts the data distribution of each domain based on the model's loss using an exponential ascent algorithm. However, dy-

namic batch loading offers a more flexible and less complex approach that can be applied to various scenarios.

Doremi follows a multi-step process: (1) Train a reference model. (2) Train a proxy model to estimate the proportion of data from each domain by adjusting the proportion based on the proxy model's loss. (3) Train the final model using the estimated data distribution. In contrast, dynamic batch loading can be directly applied to any model without the need for a reference or a proxy model. Dynamic batch loading begins by deriving a reference loss based on a fixed evaluation set. This reference loss can be estimated using scaling laws or simply by using the source model's evaluation loss. During training, the data proportion is adjusted in real-time based on the periodically measured evaluation loss. The dynamic batch loading process can be seamlessly integrated into the standard pre-training pipeline, as evaluating the loss is computationally efficient and does not introduce significant overhead. Although dynamic batch loading relies on a fixed evaluation set, which may not fully represent the model's performance on the entire dataset, this issue can be mitigated by periodically updating the evaluation set during training.

▷ **When multiple source model sizes are available, how do you choose the source model size for pruning?**

Determining the optimal source model size for pruning is challenging. However, we can perform a thought experiment by considering each parameter as a uniform "unit of information." For instance, if a source model with 7B parameters is trained using 2T tokens, we can assume that each parameter carries approximately 285 tokens of information, assuming a uniform distribution of information across the parameters. When randomly pruning this model down to 1.3B parameters, the total amount of information is reduced to $1.3B \times 285 = 0.37T$ tokens. In contrast, if we prune a 13B model (also trained with 2T tokens) down to 1.3B parameters, the total amount of information is reduced to $1.3B \times (2T / 13B) = 0.2T$ tokens. Although this estimation is rough, it suggests that pruning from a larger model may be less effective, especially when the source models are trained with the same number of tokens. It is important to note that this is a simplified estimate, and the assumption of uniform information distribution across parameters may not hold in practice. Moreover, the structured pruning process itself clearly breaks this assumption. Nonetheless, this thought experiment provides a general sense of how the source model size can impact the effectiveness of pruning.

