# OpenReview forum: "Sheared LLaMA: Accelerating Language Model Pre-training via Structured Pruning"
_ICLR.cc/2024/Conference — ICLR 2024 poster_

### Official Review · Reviewer_urdF · 2023-10-29

**Soundness:** 2 fair
**Presentation:** 3 good
**Contribution:** 2 fair
**Rating:** 6
**Confidence:** 4

**Summary:**

This paper introduces a pruning technique and a dynamic batching technique to continue training pre-trained LLMs. The proposed pruning technique offers control over the final network shape by solving a constrained optimization problem, where the shape constraint must be optimized while maximizing language modeling performance. The dynamic batching technique works by first estimating reference losses for different data domains via computing scaling curves and then adjusting weighting terms for different domains based on the difference between the reference loss and validation loss at different validation intervals. The dynamic batch loading method sounds a bit heuristic, and it requires computing scaling curves which can be often expensive (otherwise the algorithm cannot work), which may need a lot of computing and training time to obtain reference losses if there are many domains. Therefore, the efficacy of this method is questionable.

**Strengths:**

The paper is clearly written and provides enough context to understand the proposed contents. Given the simplicity of the proposed pruning technique, reproducing the results seems fairly easy. Additionally, the pre-trained checkpoints of the source LLM and the training text dataset are open-sourced, which is another plus. The proposed methods could be practically useful, as there seem to be some use cases where they can help. The pruning technique with target shape constraint shows a potential that this can work on different types of neural architecture, however, there is no empirical evidence in the paper.

**Weaknesses:**

The proposed pruning method is sound, but it is very specific for the Transformer architecture.

The title "ACCELERATING LANGUAGE MODEL PRE-TRAINING VIA STRUCTURED PRUNING" is somewhat misleading, as it suggests that the paper is proposing a generic pre-training method. My understanding is that these methods only work in a limited setting. First, a competitive pre-trained checkpoint is required. The proposed methods cannot be used when a model needs to be pre-trained from scratch. However, they may be effective in certain cases, such as when an LLM needs to be compressed and trained on different data domains (not necessarily the same ones used in the original pre-training).

Additionally, the authors' claim that this method can significantly reduce training costs compared to other LLMs trained from scratch is an overstatement, e.g. saying things like the proposed variant is outperforming baselines (which are usually trained on 300B~1T tokens) by only trained on 50B tokens. The authors should count the cost of training the source LLM as the worst-case scenario. Comparing the performance with other baselines without counting the significant training computes used for the source LLMs (7B parameters and 2T tokens) is unfair. To my knowledge, the dynamic batch loading technique is required to recover the performance on the data domains after pruning. In that case, the cost to compute the reference losses should also be reported somewhere in the paper, and it should be mentioned in the main table. It is misleading to only count the tokens used for a single training run of the pruned model when there are significant prerequisites to make it work in the first place (pre-trained and competitive LLMs and performing scaling studies on different sub-datasets).

If the same methods can also be applied to another source LLM and show similar improvements over baselines, this would reinforce the reported findings. It is difficult to distinguish whether the improvement is due to the superiority of the source LLM or the proposed methods.

Minor comments:
In Section 2.2, you wrote " pre-training dataset RedPajama (TogetherAI, 2023b; LLaMA’s pre-training dataset)". RedPajama is not the exact pre-training dataset used to train LLAMA2, but it's an open-source version trying to replicate the original training corpus, am I correct?

If Figure 8 is mentioned in the main paper, it should be included in the main paper, not the appendix.

**Questions:**

How much compute wasused to get the reference loss for each domain? Can the authors provide a breakdown and total costs?

What happens if authors continue training the model (sheared LLaMA) beyond 50B tokens?
The training cost used for sheared LLaMA is as follows: 2T (7B parameter LLM) + 50B (2.7B LLM).
If one continues training one of the baselines, e.g., Open-LLaMA-3B-v2 for an additional 1T token, would sheared LLaMA still outperform?

---

> ### Author Response · Authors · 2023-11-17
> **About the generality of the method, fair descriptions of the contribution, pruning other models**
>
> We are greatly encouraged by the reviewer's positive feedback regarding the practicality and reproducibility of our work. In response to the reviewer's comments, we provide the following responses:
>
>
> **1. Specific to Transformer architecture**
>
> Transformer architectures have become ubiquitous in both vision (ViT) and language (LLMs) models, delivering state-of-the-art results on a wide range of applications. Their widespread relevance underscores the need for specialized pruning techniques for Transformer-based models. Moreover, considering the high training costs of large-scale Transformer-based LLMs – more than any other neural network – developing efficient training and compression methods for these transformers is vital.
>
> **2. Justification for the generality of our approach**
>
> Thanks for your input! We would like to argue that our method has broader applicability for several reasons:
> - The landscape of open-source LLMs is evolving at a blazingly fast pace. In the recent two months, many strong open-source models have been released, e.g., Mistral-7B (the strongest 7B model so far), Yi-34B (the strongest 34B model so far), DeepSeek-Coder (the strongest open-source code model so far with 1.3B, 6.7B, and 33B). However, such models are usually not released across many scales, largely due to the expensive training cost. Our method, instead, provides **an efficient general approach to derive smaller-scale foundation models from any existing models**, including all the ones that I mention, to any size and configuration. We believe our work conveys an important message: to pre-train a new model, we should structurally prune from existing models instead of training it from scratch (unless the target model is larger than all the existing models).
> - The method is not limited to the strongest suite of models. In the general response, we show that pruning relatively weaker models (Pythia models) also accelerates pretraining a smaller model.
> - The method could also be useful for developing domain-specific models as the reviewer pointed out.
>
> In summary, our approach presents a compelling strategy for expediting the development of open-source large language models (LLMs) at various scales, and most importantly is affordable to communities with limited computational resources.
>
> **3. Need to take 2T pretraining data into account**
>
> Thank you for your input! Our approach is grounded in the realistic context where strong large language models (LLMs) are readily available, yet typically not in a diverse range of scales, particularly the latest releases, e.g., the strongest 7B model Mistral only has one single size. Our paper underscores the strategic use of these existing assets to derive smaller models at varied scales. While the pre-training expense of the source model is significant, it's essential to recognize that:
>
> - this often represents a sunk cost, usually borne by well-resourced institutions (as exemplified by our non-requirement to train the publicly available LLaMA-7B).
> - this cost can be effectively spread over the process of pruning for different scale models, thereby optimizing resource utilization.
>
> **4. Compute for calculating the reference loss**
>
> In terms of acquiring reference losses, we ran inference on 14M tokens with 3 models (Llama2-7b, 13b, and 70b). And the whole process took less than 4 A100 GPU hours. The cost to compute the reference losses is negligible compared to pruning (training on 0.4B tokens) and dynamic batch loading (training on 50B tokens). We have added this information to our paper.
>
> **5. Pruning another source LLM**
>
> Thanks for your suggestion! We conducted similar experiments on Pythia, and please refer to the general response, **1. Pruning a weaker and smaller foundation model**, for more details.
>
> **6. LLaMA’s pretraining data and RedPajama**
>
> Thanks for capturing this – updated in our revision!
>
> **7. Placement of figure 8**
>
> Thanks for the comment and we updated the draft!

---

> ### Author Response · Authors · 2023-11-17
> **About continuing pre-training a small model**
>
> **8. Continue pre-training a small pre-trained model**
>
> Thanks for your insightful question! Our study was constrained to a 50B token budget due to computational limitations. Nevertheless, as depicted in Figure 1 of our paper, the trend of continued pre-training reveals no sign of convergence in performance. This strongly suggests that increasing the data scale further would continue to enhance the model."
>
> Regarding your second question, further finetuning Open-LLaMA-3B-v2 for an additional 1T tokens was beyond our experimental capacity. Instead, we focused on smaller-scale Pythia models, each trained with 300B tokens. We pruned the Pythia-410M model down to 160M and continued its pre-training with 33B tokens (Sheared-Pythia-160M). This was compared against continuing the pre-training of Pythia-160M with an equivalent token amount. The revised Figure 11 shows that while Pythia-160M (300B+33B) initially outperforms, Sheared-Pythia-160M (300B+33B) learns more rapidly, eventually surpassing Pythia-160M. These preliminary results hint at a potential for greater performance gains from Sheared-Pythia with increased computational resources.
>
> We think that pruning a larger model would yield more substantial benefits. This could be supported by findings from a previous study [1], which demonstrated that larger models perform better when pruned than training small models from scratch because larger models are easier to optimize. However, we will defer a deeper analysis of this hypothesis on the LLM scale to future work!
>
> [1] Train Large, Then Compress: Rethinking Model Size for Efficient Training and Inference of Transformers

---

> ### Author Response · Authors · 2023-11-23
>
> Dear Reviewer urdF,
>
> We’d love to hear your thoughts on our newly added results and hopefully they have addressed your concerns for the paper! We really appreciate your efforts in providing insightful comments and suggestions to us!

---

### Official Review · Reviewer_bT2X · 2023-10-31

**Soundness:** 3 good
**Presentation:** 2 fair
**Contribution:** 3 good
**Rating:** 5
**Confidence:** 4

**Summary:**

This paper introduces Sheared LLaMA, a method focused on developing streamlined LLaMA models through the pruning of pre-trained ones. Central to this approach is a sparse training technique that employs L0 regularization to systematically zero out specific substructures. Additionally, the paper presents a dynamic batch-loading strategy, effectively recalibrating the significance of different data domain. In the fine-tuning phase, Sheared LLaMA undergoes training on 50 billion tokens, ultimately attaining a performance level comparable to models produced by scratch training.

**Strengths:**

* This work studies a practical approach to craft lightweight LLaMA by pruning, which requires less training cost.
* The proposed model achieves superior performance compared to publicly available models.
* The dynamic batch loading is interesting.

**Weaknesses:**

My main concern lies in the technical novelty and effectiveness of the proposed pruning and sampling method. And it's unclear which part plays the most important role in pruning & fine-tuning.

* Sheared LLaMA employs regularized training as a preliminary step before pruning. However, **the effectiveness of the proposed constrained optimization remains unclear**. As illustrated in Table 10, the performance gap between Sheared LLaMA and LLM-Pruner (a simple Taylor-based method) is marginal (ΔPPL=0.24). Questions arise regarding whether this improvement is attributable to the proposed pruning method or the dynamic batch loading. Additionally, it's unclear if the LLM-Pruner baseline was also trained with dynamic batch loading.
* Based on the previous questions, this work mentioned that "This observation indicates that pruning preserves a greater amount of knowledge in low-entropy and smaller domains (e.g., GitHub) compared to high-entropy and larger domains (e.g., C4)". However, this phenomenon might be also caused by a biased regularization before pruning, since the regularized training step only saw 0.4B tokens as mentioned in Sec. (4.3). Maybe this problem can be avoided by sampling balanced data from different domains for regularization, or deploying a simple Taylor expansion [1, 2] with balanced data.
* Table 4 presents some confusing results: CoFiPruning [1] shows superior Perplexity (PPL) but inferior Throughput compared to Sheared LLaMA. This raises the question: are CoFiPruning and Sheared LLaMA comparable in performance? Given these mixed results, it's difficult to conclusively state that Sheared LLaMA has outperformed CoFiPruning.
* Might be a typo: There's an inconsistency in reporting the size of the LLM-Pruner baseline -- labeled as 1.3B in the figure caption but noted as 1.6B in the table. Clarifying these comparisons would be helpful.


[1] Xia, M., Zhong, Z., & Chen, D. (2022). Structured pruning learns compact and accurate models. arXiv preprint arXiv:2204.00408.
[2] Molchanov, P., Mallya, A., Tyree, S., Frosio, I., & Kautz, J. (2019). Importance estimation for neural network pruning. In Proceedings of the IEEE/CVF conference on computer vision and pattern recognition (pp. 11264-11272).
[3] LeCun, Y., Denker, J., & Solla, S. (1989). Optimal brain damage. Advances in neural information processing systems, 2.

**Questions:**

Please refer to the weaknesses.

---

> ### Author Response · Authors · 2023-11-17
> **About a fair comparison to LLM-Pruner, why dynamic batch loading, comparison to CoFiPruning**
>
> Thank you for the valuable feedback! We appreciate that you acknowledge the efficiency implications of our proposed approach to pre-training and the strong performance of our models. We address your comments below:
>
> **1. A far comparison to LLM-Pruner**
>
> We compared our pruning approach to LLM-Pruner, both without dynamic batch loading. Please refer to the general response, ** A fair comparison compared to LLM-Pruner**, where we provided a fair and thorough comparison to LLM-Pruner in terms of loss, model architecture, inference speed and training throughput. Besides, the PPL difference of 0.24 roughly corresponds to 1 point difference in downstream performance on the training trajectory, and we do not consider it as marginal.
>
>
> **2. Dynamic batch loading**
>
> The core contribution of dynamic batch loading is exactly to automatically figure out **what the “balanced data”** should be for both pruning and continued pre-training, without adding additional compute overhead.To add additional contexts:
> In our initial experiments, we also explored directly excluding “easy domains” like GitHub, StackExchange, and ArXiv during the pruning stage. However, it does not address the loss imbalance issue. Please refer to our response to reviewer qt4Y if you are interested in more details.
> We also explored pruning with more tokens but generally observed that the loss imbalance issue persists. Hence the loss imbalance problem is not likely to be solved by putting in more pruning budget.
> [1, 2] are about different pruning techniques. We experimented with [1], and still observe a similar issue with loss disparity across domains.
>
> In conclusion, we argue that such a loss imbalance problem cannot be easily solved by manually curating data, adding more pruning budgets, or switching pruning methods. We believe our dynamic batch loading is essential, and its strength is demonstrated in our ablations in Section 4.1.
>
> **3. Comparison to CoFiPruning**
>
> We recognize that Sheared-LLaMA does not outperform CoFiPruning in perplexity when both methods use the same amount of data for pruning. However, a key difference is that CoFiPruning produces a non-uniform model, which introduces extra inference costs.
>
> In practice, the perplexity gap could be compensated by slightly training Sheared-LLaMA longer for about 0.6B tokens in our experiment. Since this extra training is a one-time cost and inference speed is a crucial factor for LLM serving, the choice to prune towards a uniformly layered architecture is justified for long-term efficiency.
>
> **4. Typo**
>
> Thank you for catching the typo; we've updated the paper. The LLM-Pruner has 1.6B parameters, 0.3B more than Sheared-LLaMA due to its lack of support for pruning hidden dimensions, including word embeddings. While embeddings have minimal inference overhead, we've maintained parity in the remaining model parameters between Sheared-LLaMA and LLM-Pruner for a fair comparison.

---

> ### Author Response · Authors · 2023-11-23
>
> Dear Reviewer bT2X,
>
> We hope our newly added experiments help address your concerns for the paper! We would love to hear your thoughts about them and provide more information if needed.
>
> Really appreciate it!

---

### Official Review · Reviewer_qt4Y · 2023-11-01

**Soundness:** 4 excellent
**Presentation:** 3 good
**Contribution:** 4 excellent
**Rating:** 8
**Confidence:** 4

**Summary:**

The manuscript introduces a technique that prunes pretrained models towards a target (smaller) architecture rather than towards a target sparsity level. After a small amount of retraining, the pruned models outperform similar-size models trained from scratch on many more tokens than the retraining budget, suggesting that the pruning approach is a more efficient path to producing a small model than training from scratch. The efficiency of retraining is enhanced by a simple and effective data selection scheme. Comparisons with other pruning methods show that "targeted structured pruning" (i.e., "shearing") can lead to models that are faster than those created by other approaches.

**Strengths:**

The manuscript focuses on the problem of creating a compact LLM -- existing approaches either train from scratch (which is expensive and forgoes inheriting knowledge from stronger/larger models) or prune an existing model (which can lead to a suboptimal structure for inference speed). By pruning pretrained models to a target architecture with targeted structured pruning and using a short retraining period ("continued training"), the submission's proposed approach is able to address the prior problems in this significant area. The resulting models are trained faster and perform better than models trained from scratch, and they perform inference faster than models pruned by other approaches.

The shearing algorithm ("targeted structured pruning") is very clearly explained (with nice illustrations) and is simple despite its flexibility and power. The idea of pruning towards a target architecture is (as far as I know) novel.

To enhance learning in the pruned model during "continued training", an original scheme to dynamically change the data mixture is developed ("Dynamic Batch Loading"). This scheme is not only interesting for its usefulness to Sheared-LLaMA -- as the authors suggest, it could be used to help make any model's training more efficient by avoiding usage of training data that makes relatively little progress towards the desired model performances.

The analyses (e.g., of Dynamic Batch Loading, of other pruning approaches, etc.) thoroughly support the manuscript's arguments.

**Weaknesses:**

While the paper is very thorough, additional inference timings for models produced by other competitive methods could help readers better understand the merits of the proposed approach (see "Questions" below).

**Questions:**

Score-affecting:

1. Expanding Table 4 with inference speeds for the following models could help readers better understand the importance of the Sheared-LLaMA targeted pruning approach.
   - Before shearing
   - Before shearing with 2:4 sparsity and with 4:8 sparsity. This would allow us to compare Sheared-LLaMA to both older and newer pruning approaches (like Wanda).
   - LLM-Pruner model

Interesting:

1. To better understand the effect of the discovered pruning mask on performance during the "continued training" period, perhaps you could use only C4 data to find the pruning mask. If you did this, would "continued training" on the mixture of domains (without dynamic batch loading) still show GitHub performance reaching its reference loss and C4 performance not reaching its reference loss? Some related questions follow.
   - Are losses being reduced at different rates across domains because the pruning mask was learned on too little data to account for the complexity of some domains (like C4)?
   - 2 million tokens (500 sequences) are used for each held-out set: should the held-out set size be larger for more complex datasets (e.g., C4) to ensure that the pruning mask is found on a representative set of data?

2. A version of Figure 4 with perplexity or loss.

Minor:

1. There are numerous typos. Please proofread the paper carefully. Some examples follow:
   - "Sheared-LLaMA-3B" is mentioned but 2.7B is probably intended.
   - $z^{inter}$ and $z^{int}$ are used interchangeably, so are $H^T$ and $H_T$.
   - Table 2 seems to have NQ and LAMBADA switched in the 7B row.
   - Section 4.1: "hypothetical 2.7B parameter LLaMA2 model" is stated. Do you mean 1.3B?
   - Figure 8's caption is wrong.

2. How are the Lagrange multipliers initialized? Relatedly, a reference for the specific Lagrange multiplier approach used might be nice to include -- Platt and Barr (1987) looks related but not exactly the same.

3. In section 4.2, the sentence "non-uniformity also introduces training and inference overhead due to irregularities in model architectures" could be followed by intuition/clarification that explains why irregularities add overhead.

4. In Table 4, consider adding perplexity of Sheared-LLaMA *with* continued pretraining to complement the *without* continued pretraining numbers.

5. Like Kaddour et al. (2023), "Compute-Efficient Deep Learning" (Bartoldson et al., 2023) shows why promised gains "may not be consistently realized" -- it also discusses (in its survey) all of the various efficient training approaches mentioned in the submission's Related Work section.

---

> ### Author Response · Authors · 2023-11-17
> **Inference speed, pruning with C4**
>
> **1. Adding inference speedup for different pruning approaches**
>
> Thanks for your suggestion. We've included below the inference speed (tokens/second) comparisons for all methods mentioned you mentioned. These experiments were all conducted on a single A100 80GB GPU, generating up to 2048 tokens with a batch size of 1.
> | Model         |  7B|
> |---------------|------|
> | LLaMA2 | 37 |
>
>
> | Model         | 1.3B | 2.7B |
> |---------------|------|------|
> | LLM-Pruner    |  41  |  40  |
> | Sheared-LLaMA |  62  |  47  |
>
> | Model         | 50% sparsity |
> |---------------|------|
> | Wanda (2:4)   |   42  |
> | Wanda (4:8)   |   42  |
>
> Sheared-LLaMA is way more efficient at inference than LLM-Pruner, largely due to the better chose final architecture (See general response,  ** A fair comparison compared to LLM-Pruner**, for a more thorough comparison). Wanda type of semi-structured pruning also achieves inference speedup compared to the source model. But it is not as fast as small dense models, and is restricted to 50% sparsity. Measuring throughputs (using the maximum batch size) will give Sheared-LLaMA even more advantage than this setup. We have added the analysis to Appendix H.
>
> **2. Pruning with CC, C4, Wiki and Books**
>
> Regarding point 1, in the project's initial phase, we experimented by pruning without including data from GitHub, StackExchange, and ArXiv (domains retaining more performance). We specifically pruned LLaMA1-13B to 7B using a mix of C4 (40%), CC (40%), Wiki (10%), and Books (40%) datasets. Post-pruning, we further pre-trained this model using the RedPajama dataset with all of the domains.
>
> As shown in the table below, post-pruning perplexity differences were fairly uniform across domains. However, after continued pre-training with data from all the domains of RedPajama, the loss disparity grew: the loss difference on GitHub/StackExchange got much smaller than those on domains like C4. This indicates that **simply excluding easier-to-recover domains during pruning** doesn't address the loss disparity issue in continued pre-training.
>
> These findings led us to develop dynamic batch loading, a more effective solution to tackle domain loss disparity. For a comprehensive understanding, we've detailed these initial experiments in Appendix G of our revised submission.
>
> |                           |    CC   |  GitHub |  Book  | StackExchange | Wikipedia |  ArXiv |   C4   |
> |---------------------------|:-------:|:-------:|:------:|:-------------:|:---------:|:------:|:------:|
> | llama-13b                 |  1.7585 |  0.6673 | 1.9499 |     1.4207    |   1.4331  | 1.3855 | 1.8619 |
> | llama-7b                  |  1.8366 |  0.7108 | 2.0322 |     1.5112    |   1.5291  | 1.4340 | 1.9331 |
> | Pruned model (w/o Github) |  2.1849 |  1.0971 | 2.3726 |     1.9080    |   2.1151  | 1.7542 | 2.3187 |
> | diff from llama-7b        |  0.3483 |  0.3863 | 0.3404 |     0.3968    |   0.5860  | 0.3202 | 0.3857 |
> | Continue Pretrain (w RP)  |  1.8344 |  0.6325 | 2.0984 |     1.4542    |   1.4549  | 1.4460 | 2.0395 |
> | diff from llama-7b        | -0.0022 | -0.0783 | 0.0661 |    -0.0570    |  -0.0743  | 0.0120 | 0.1064 |
>
> **3. Use a larger set of validation data for more complex domains**
>
> Enlarging the validation set of complex domains for training stability is an excellent idea (point 2), which will be likely to provide improved calibration and stability for dynamic data loading. We did not explore this direction thoroughly in our initial experiments, so there is certainly room to build on this suggestion in future work to better understand the impacts!

---

> ### Author Response · Authors · 2023-11-17
> **More interesting analysis**
>
> **4. A version of Figure 4 with perplexity or loss.**
>
> In our revised draft, we added Figure 12 to illustrate the loss trends during continued pre-training of both the INCITE model and our pruned model. Notably, the INCITE model's loss remains relatively unchanged throughout this phase, whereas the loss of our pruned model progressively decreases, ultimately falling below that of the INCITE model. This trend is consistent with the downstream task performance depicted in Figure 4.
>
> **5. Typos**
>
> Thanks for spotting them our typos and we have fixed all of them in the revision.
>
> **6. Initialization of Lagrange multipliers**
>
> We randomly initialize the Lagrange multipliers with value 0. Additionally, we've incorporated the related work you mentioned – It’s nice to get to connect to classical works!
>
> **7. Explanation for non uniformity**
>
> Thanks for pointing this out! The GPU architecture and model parallelism used in training large models mean that non-uniform structures can increase memory allocation overhead and disrupt load balancing across GPUs. Additionally, varying shapes, even with the same number of elements, can significantly impact inference latency for matrix multiplication operations. We empirically demonstrate in Section 4.2 that uniform pruned structures outperform non-uniform ones in speed. We added these intuitive explanations to our revised draft.
>
> **8. adding perplexity after continued pretraining**
>
> For a fair comparison, we only included the pruned models’ perplexities without continued pre-training in Table 4. In the future, we will consider adding an experiment on continuing pre-training CoFiPruning to complete this comparison. Thanks for the suggestion!
>
> **9. Adding Bartoldson et al., 2023**
>
> Thanks for the nice pointer! We have incorporated it into our revision.

---

> > ### Comment · Reviewer_qt4Y · 2023-11-23
> >
> > These extra results are great, thank you for adding them!
> >
> > My concerns have been addressed.

---

### Official Review · Reviewer_wgpN · 2023-11-01

**Soundness:** 3 good
**Presentation:** 3 good
**Contribution:** 2 fair
**Rating:** 5
**Confidence:** 4

**Summary:**

This paper introduces to employ the structural pruning to reduce the pre-training cost of LLMs.  It focuses on the integration of two techniques: targeted structured pruning and dynamic batch loading. This paper proposes two small models: Sheared-LLaMA-1.3B and Sheared-LLaMA-2.7B. Comparative testing reveals that these lightweight models are capable of outperforming other counterparts with 1.3B and 2.7B parameters.

**Strengths:**

1. This paper introduces two scaled-down versions of LLaMA-2, achieved through structured pruning.
2. Experiments demonstrate that these two small LLMs exhibit superior performance when compared to other models such as OPT, Pythia, INCITE, and Open-LLaMA.
3. The performance and capabilities of these two pruned models are extensively assessed using the Open LLM Leaderboard.

**Weaknesses:**

1. My main concern centers around the novelty of this paper. The first method, named targeted structured pruning, has been previously employed in several papers, with some offering further advanced variations [1, 2, 3, 4, 5]. As for the second method, dynamic batch loading, it closely resembles the application of the technique from [6]. The main observation by the authors that structured pruning can reduce training costs is a well-known advantage of all the structured pruning methods.

2. The paper lacks experiments to show the effectiveness of targeted structured pruning. No experiments can be found to verify the effectiveness of the 'targeted structured pruning` compared with other pruning algorithms. Given that the proposed targeted structured pruning falls within the realm of structural pruning, a comparative analysis with LLM-Pruner[7] is essential to establish whether the newly proposed method improves upon the existing techniques for structured pruning of large language models. Utilizing the same experimental settings as LLM-Pruner for this comparison would provide a more direct and clear demonstration of the proposed method's effectiveness.

3. I cannot tell whether the enhanced performance is attributed to a stronger foundational model (LLaMA2) from which pruning occurs. It's conceivable that starting with a stronger base model could lead to better results post-pruning. Have any experiments been conducted where a model, for instance, OPT-2.7B, is pruned down to 1.3B and then compared against the officially pre-trained OPT-1.3B? (Given the limited time available for rebuttal, it might not be feasible to pre-train a 1.3B model. Conducting a similar experiment with a considerably smaller model, such as a 350M version, could also provide valuable insights.)

[1] Learning Sparse Neural Networks Though L0 Regularization. 2017.
[2] Learning Structured Sparsity in Deep Neural Networks. 2016.
[3] Neural Pruning via Growing Regularization. 2020.
[4] Fast Post-Training Pruning Framework for Transformers. 2022.
[5] Dynamic Sparse Training: Find Efficient Sparse Network From Scratch With Trainable Masked Layers. 2020.
[6] DoReMi: Optimizing Data Mixtures Speeds Up Language Model Pretraining. 2023.
[7] LLM-Pruner: On the Structural Pruning of Large Language Models. 2023.

**Questions:**

1. Why the dynamic batch loading is a more efficient one than Doremi?

---

> ### Author Response · Authors · 2023-11-17
> **About novelty, comparison to LLM-Pruner, and pruning another source model**
>
> We are encouraged that the reviewer acknowledged that the Sheared-LLaMA models underwent thorough evaluations and demonstrated strong performance. Thanks for your comments and suggestions on the work!
>
> **1. Concerns about the novelty of the paper.**
>
> We disagree with this point, as our proposed methods of targeted structured pruning and dynamic batch loading are both different from previous work significantly, which we clarify below:
>
> **Targeted structured pruning:**
> - The goal of `targeted structured pruning` is to achieve an **exact target transformer architecture** that is pre-specified based on a source transformer model. To the best of our knowledge, none of the works mentioned by the reviewer [1, 2, 3, 4, 5] have done this. This is based on the intuition that existing models’ configurations have already been well optimized to balance model expressivity and inference efficiency, and empirically we showed that it is more effective than pruning to non-uniform structures with only sparsity specified.
> - Another important contribution is that we prune a general-purpose, large language model with billions of parameters (7 billion parameters in our experiments), **a scale that was not demonstrated in the literature**. The implications of pruning at such a large scale are not well understood, and our work serves as one to provide such insights.
>
> **Dynamicbatch loading:**
> - Although our dynamic loading method is inspired by Doremi [6], they differ substantially in that
> Doremi requires a complicated multi-staged process to train a reference model and a proxy model before the final training run. Our dynamic batch-loading algorithm does not require training any additional models, making it **significantly more efficient** and easier to use than Doremi.
> - The other contribution is that we introduce a reference loss derived from an estimate of scaling law, which takes scale differences into account, and is a more principled solution to estimating a reference loss.
>
> We have updated the paper to better highlight our contributions and differences with prior works in Section 2.1 and 2.2.
>
>
> **2. Fair comparison to LLM-Pruner**
>
> Please refer to the general response, **1.A fair comparison compared to LLM-Pruner**, where we provided a fair and thorough comparison to LLM-Pruner in terms of loss, model architecture, inference speed and training throughput.
>
> **3. Pruning a weaker and smaller foundation model**
>
> Thanks for your suggestion! We conducted additional experiments on Pythia-440M models, and please refer to the general response, **2. Pruning a weaker foundation model**, for more details.

---

> ### Author Response · Authors · 2023-11-23
>
> Dear Reviewer wgpN,
>
> As it’s the last day for discussion, we wonder if our responses have addressed your concerns for the paper. We would really love to have your inputs on it and are happy to provide more information if needed!

---

### Author Response · Authors · 2023-11-17
**General Response**

We thank all reviewers for their constructive feedback! We have updated our paper to incorporate suggestions on typos/clarifications and added the following results:

**1. A fair comparison compared to LLM-Pruner**

Reviewers wgpN and bT2X highlighted the need for a fair comparison with LLM-Pruner without dynamic data loading. We confirm that both our proposed pruning algorithm and LLM-Pruner used original data proportions of RedPajama without dynamic data loading in Section 4.2 and Appendix E.2. For clarity, these sections have been updated with detailed comparisons.

For the following results, we compare our pruned model with LLM-Pruner, with both having the same number of parameters (1.23B; embeddings excluded) for a fair comparison.

In terms of losses, our targeted structured pruning **without dynamic batch loading** reaches a lower loss than LLM-Pruner with the same data and compute invested (Shown in Figure 8).

In terms of model architectures, the LLM-Pruner model leads to a configuration where intermediate layers are smaller than the hidden dimensions, deviating from the typical Transformer design where intermediate sizes are typically 3-4 times the hidden size. This limitation stems from LLM-Pruner's inability to prune layers and hidden dimensions effectively, resulting in suboptimal architecture at higher sparsity levels (the original study only investigates up to 20% sparsity).

Regarding training/inference throughput, our pruned model structure has a substantial speed advantage compared to LLM-Pruner, as shown in the table below. For inference speed analysis, we use a single A100 GPU to generate up to 2048 tokens. For training speed analysis, we use 16 A100 GPUs to do language model pre-training with a length of up to 2048.

|            | Inference Speed | Training Throughput | PPL |
|------------|:----------:|:----------:|:----:|
| LLM-Pruner |     41 tokens/s     | 83K tokens/s     |7.09 |
| Ours       |     62 tokens/s    | 139K tokens/s    |6.85 |




**2. Pruning a weaker foundation model**

Both reviewers wgpN and urdF suggested we should prune a weaker foundation model to understand the efficacy of the proposed approach.

This is an excellent point. We actually began this project working on Pythia models [1], which is a series of pre-trained LLMs at different scales with open-source data/checkpoints. They are generally stronger than OPT [2].

Concretely, we pruned a 440M Pythia model down to 160M using its pre-training data, the Pile [3]. We then continued pre-training the pruned 160M model on 33B additional tokens with dynamic batch loading (also on the Pile). Below, we show the average 0-shot performance on ARCC, ARCE, Lambada, LogiQA, PIQA, Sciq, and Winograd. **The Sheared-Pythia model outperforms the 160M Pythia by 2 points** on average. Impressively, Sheared-Pythia-160M achieves the same performance level as Pythia-160M with just 10B tokens, which is only 3% of the original pre-training cost.

This set of experiments demonstrates the generality of our pruning solution to effectively produce smaller-scale models more efficiently. We have incorporated this set of experiments in an additional section in Appendix E.5 in the updated draft.

[1] Biderman et al., 2023. Pythia: A Suite for Analyzing Large Language Models Across Training and Scaling
[2] Zhang et al., 2022. OPT: Open Pre-trained Transformer Language Models
[3] Gao et al., 2020. The Pile: An 800GB Dataset of Diverse Text for Language Modeling.

|                | Tokens | Performance |
|----------------|:-----------:|:-----------:|
| Pythia-160M    |               300B               |    43.56    |
| Sheared-Pythia-160M |            (300B) 10B              |    43.51    |
| Sheared-Pythia-160M |             (300B) 33B              |    45.87    |

---

### Author Response · Authors · 2023-11-20
**Reminder of the discussion period deadline**

We'd like to extend our gratitude to the reviewers again for your insightful feedbacks! Your recognition of strong empirical results and potential implications of the proposed approach has been incredibly encouraging.

As the deadline of the discussion period approaches, we'd like to kindly remind the reviewers to examine the new supporting results we added as follows and share your thoughts with us!

- enhancing clarity of the novelty of the work
- pruning another foundational model
- comparing with the baseline fairly
- analyzing inference latency
- and more!

Authors 4531

---

### Meta-Review · Area_Chair_zTU2 · 2023-12-12

**Metareview:**

This paper introduces a pruning technique and a dynamic batching technique to continue training pre-trained LLMs. It received scores of 5568, which is quite divergent.

On the one hand, the reviewer who gave a score of 8 commented that the shearing algorithm ("targeted structured pruning") is very clearly explained (with nice illustrations) and is simple despite its flexibility and power. The idea of pruning towards a target architecture is (as far as I know) novel. On the other hand, other reviewers questioned the novelty of the method, and asked for fair comparisons to LLM-Pruner.

The AC thinks that the authors have done a nice job during rebuttal, and overall, the merits outweigh the flaws, therefore, would like to recommend acceptance by the end.

**Justification For Why Not Higher Score:**

Some reviewers questioned the novelty of the method.

**Justification For Why Not Lower Score:**

Overall, the paper is well written, experiments are sufficient, and one reviewer found this paper especially inspiring.

---

### Decision · Program_Chairs · 2024-01-16

Accept (poster)